# Two opposing hippocampus to prefrontal cortex pathways for the control of approach and avoidance behaviour

Candela Sánchez-Bellot[1], Rawan AlSubaie[1,2], Karyna Mishchanchuk [1,2], Ryan W. S. Wee[1] & Andrew F. MacAskill [1✉]

The decision to either approach or avoid a potentially threatening environment is thought to rely upon the coordinated activity of heterogeneous neural populations in the hippocampus and prefrontal cortex (PFC). However, how this circuitry is organized to flexibly promote both approach or avoidance at different times has remained elusive. Here, we show that the hippocampal projection to PFC is composed of two parallel circuits located in the superficial or deep pyramidal layers of the CA1/subiculum border. These circuits have unique upstream and downstream connectivity, and are differentially active during approach and avoidance behaviour. The superficial population is preferentially connected to widespread PFC inhibitory interneurons, and its activation promotes exploration; while the deep circuit is connected to PFC pyramidal neurons and fast spiking interneurons, and its activation promotes avoidance. Together this provides a mechanism for regulation of behaviour during approach avoidance conflict: through two specialized, parallel circuits that allow bidirectional hippocampal control of PFC.

[1] Department of Neuroscience, Physiology and Pharmacology, University College London, Gower St, London WC1E 6BT, UK. [2]These authors contributed equally: Rawan AlSubaie, Karyna Mishchanchuk. ✉email: a.macaskill@ucl.ac.uk

The decision to explore a novel or potentially threatening environment is essential for survival. Too little exploration reduces the likelihood of finding sources of reward, while too much increases risk such as injury. Dysfunction in such decision making, such as in the overestimation of threat, is thought to be a core feature of a number of anxiety disorders, and can lead to maladaptive avoidance behaviour[1]. However the neural circuitry underlying such decisions has remained elusive.

Resolving the conflict between approach and avoidance behaviour is thought to crucially depend on the activity of the ventral hippocampus (vH)[1–3]. For example, in the elevated plus maze (EPM)—a commonly used assay of innate approach avoidance conflict—the decision to explore the innately threatening open arms, or to remain in the relative safety of the closed arms is thought to be defined by the overall level of vH activity[1,3–5]. However, how this is achieved is unknown, as vH activity can both promote or inhibit approach and avoidance behaviour, dependent on how vH circuitry is manipulated[4,6–11]. Consistent with this bidirectional role, neuronal activity in vH during approach and avoidance is heterogeneous. In the EPM, different neurons in the CA1 and subiculum region of vH fire upon entry into either the open arms or the closed arms of the maze[4,12], suggesting that different populations of neurons may be involved in the promotion of either approach or avoidance behaviour.

The organization of the vH circuit is also heterogeneous, and is composed of distinct, nonoverlapping subpopulations of neurons that vary in their morphology, physiology, and local and long-range connectivity[13,14]. In particular, the downstream projection target of vH neurons is frequently used to distinguish functional specializations during behaviour[4,6,10,12,15,16]. Neurons in vH that preferentially fire during either approach or avoidance are both enriched in a subpopulation of CA1 and subiculum neurons that project to the prefrontal cortex (PFC)[12]. Notably, the neurons that make up this vH-PFC projection are markedly heterogeneous[17], further suggesting that this projection may be made up of multiple populations of neurons with distinct functions. However, how this heterogeneity relates to function during behaviour, and how this function is achieved via projections to PFC circuitry is unknown.

One way this may occur is via distinct subpopulations of neurons in vH differentially connecting to excitatory and inhibitory circuitry in PFC. Neurons in PFC track exploration in the EPM[9,18], and PFC activity defines behaviour within the maze: increased excitatory activity in PFC increases avoidance behaviour[19–21], while PFC inhibition promotes approach[22,23]. Importantly, the extent of excitatory and inhibitory drive in PFC that results from hippocampal activation is different at unique points during behaviour[24]. Notably, during approach-avoidance conflict, the influence of vH input can be either inhibitory[25] or excitatory[9] dependent on ongoing behaviour. Thus, the relative level of excitation and inhibition within PFC can be bidirectionally defined by vH, and is well placed to control the balance of approach and avoidance behaviour. However, how heterogeneous populations of neurons in vH support such flexible, bidirectional modulation of PFC is unknown.

Here we use a combination of in vitro and in vivo circuit analysis to show that the vH-PFC projection is made up of two populations of neurons distributed across the radial hippocampal axis at the CA1/subiculum border. The activity of these two populations is controlled by specialized local and long-range afferent input, and each has unique connectivity in PFC. The superficially located population promotes exploration of the open arms of the EPM, and is preferentially connected to widespread inhibitory circuitry within PFC. The deep population promotes entry to the closed arms of the EPM, and is preferentially connected to excitatory pyramidal neurons and fast spiking interneurons. Together, our data support a model where two separate vH-PFC projections bidirectionally control behaviour during approach avoidance conflict.

## Results

**The hippocampal projection to prefrontal cortex consists of two populations distributed across the radial axis.** Despite intensive investigation of the functional and behavioural properties of the vH to PFC projection, the cellular distribution of these neurons in hippocampus remains unclear. This is compounded by that fact that the precise boundary between the classical long-range output regions of hippocampus (distal CA1, proximal and distal subiculum) is unclear and widely debated[16]. To investigate the cellular organization of the projection from vH to PFC, we labelled neurons that project to the PFC with an injection of the retrograde tracer cholera toxin (CTXβ, Fig. 1a), and examined the distribution of fluorescently labelled neurons in vH. We made transverse slices of the vH to maintain the trisynaptic circuit layout, and to more easily identify the distribution of neurons across the different hippocampal subfields. Consistent with previous work[16,26], labelled neurons in these slices were located as a distribution, spread across the extent of the CA1/subiculum border (Fig. 1b). However, in contrast to the graded distribution across the proximal:distal axis, we noticed that the distribution of labelled neurons was not uniform within the pyramidal layers, and consistently formed two distinct clusters distributed across the radial axis (equivalent to the most superficial and deep regions of the pyramidal cell layer (Fig. 1b, c)).

We next confirmed this observation using unsupervised Gaussian Mixture Modelling (GMM), where the spatial distribution of vH-PFC neurons was consistently best described as two layers separated across the radial axis (Fig. 1d, e). Together this suggests that the vH-PFC projection is distributed as a continuum across the CA1/subiculum border, but is segregated along the radial axis of vH.

Previously, the distribution of projection neurons in the hippocampus has been defined using standardized atlases such as the Allen Brain Atlas[17,27,28] and the Hippocampal Gene Expression Atlas (HGEA[29]). We next wanted to compare the distribution of vH-PFC neurons to these standard atlases. Therefore, we repeated our retrograde labelling experiment, prepared coronal sections of the entire extent of the hippocampus, and registered the location of labelled neurons to the Allen Brain Atlas (ABA) coordinate framework[16,27,30] (Fig. 1f, g). This allowed us to quantify the location of vH-PFC neurons across all three anatomical axes in standard reference space, and also their relative location in either CA1 or subiculum. Notably, and consistent with a gradient-like organization of vH output[16,26], labelled neurons were distributed across the entire area around the border of CA1 and proximal subiculum as defined by the ABA, with no obvious separation between the two regions (Fig. 1g, Supplementary Fig. 1a). This was such that approximately half of the labelled neurons were present in both CA1 and subiculum (Fig. 1h). However, it should be noted that the border between CA1 and subiculum in vH is contentious and can be dramatically different across studies and atlases (see discussion and Supplementary Fig. 1).

Consistent with our transverse sections, we noticed that in many coronal sections the distribution of labelled neurons was not uniform within the pyramidal layers, and again formed two distinct clusters distributed across the radial axis (Fig. 1g). Previously, the distribution of projection neurons in the subiculum has been classified into distinct layers as part of the HGEA[29]. We therefore aligned registered vH-PFC neurons to this atlas to investigate if the two vH-PFC populations may

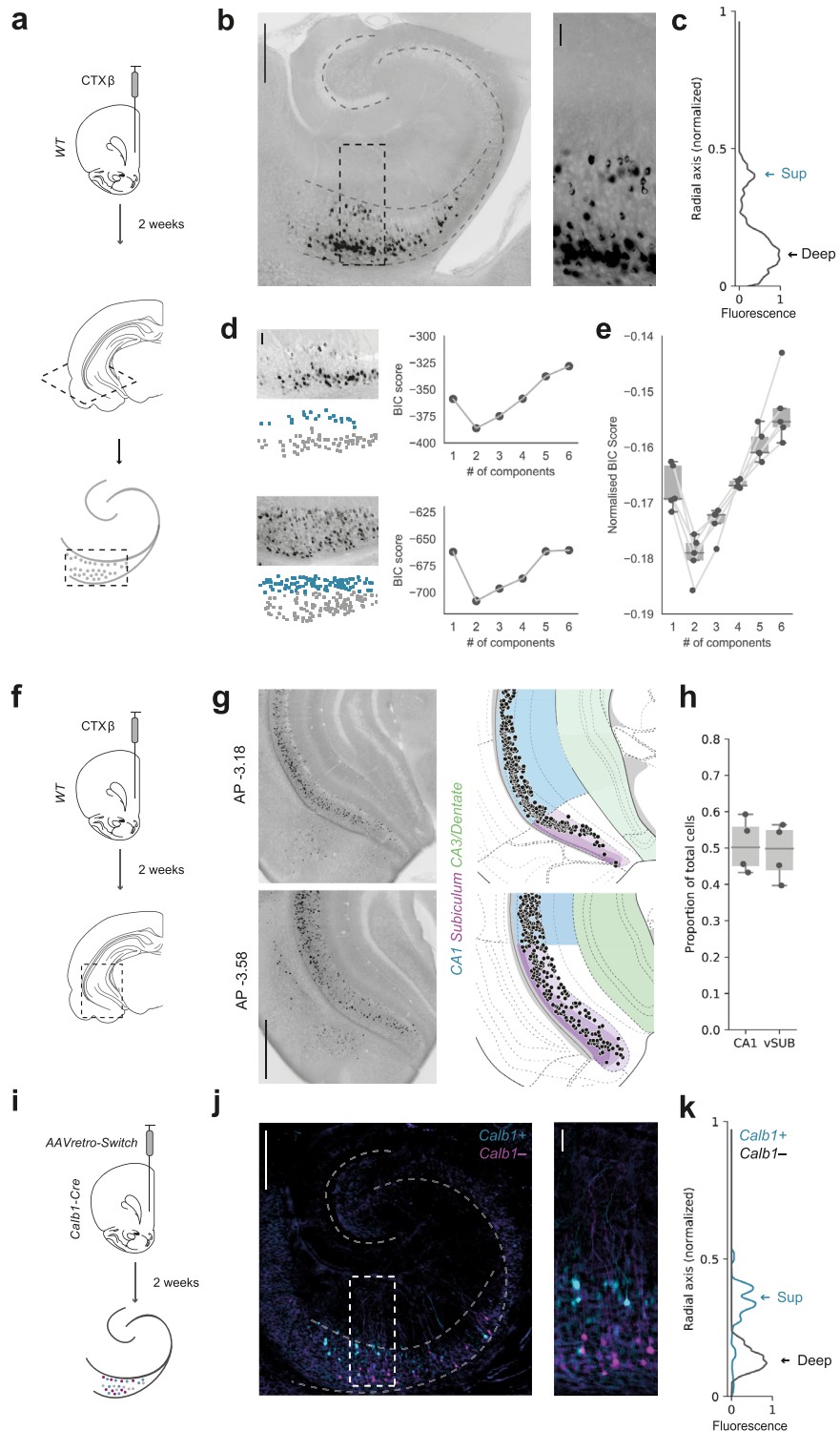

align with these previously defined layers (Fig. 1g, Supplementary Fig. 1a). We found that in subiculum the location of the two layers of vH-PFC neurons did not precisely map onto the layers defined in the HGEA. While the superficial layer corresponded to layer II, the deep layer was located at the border of layer III and IV. These results suggest that vH-PFC neurons are spread proximo:distally across the distal CA1/ proximal subiculum border, but form two layers across the radial axis, that in subiculum correspond to superficial (layer II) and deep (layer III/IV) subiculum layers.

The ventral CA1/subiculum border is classically described as having parallel output pathways, where each neuron projects to only one downstream area. However, there is also a small, yet consistent proportion of neurons that send collateral projections to multiple areas[15,16,31]. To investigate whether the two vH-PFC layers contained different proportions of these dual projecting neurons, we injected fluorescent choleratoxin into either PFC and nucleus accumbens (NAc) or PFC and lateral hypothalamus (LH). First, we investigated overall colocalization of tracers – which is suggestive of collateral projections to both regions

**Fig. 1 Hippocampal neurons projecting to PFC form two populations segregated across the radial axis. a** Schematic of cholera toxin (CTXβ) injection into PFC and retrograde labelling in transverse hippocampal slices. **b** Left, Transverse slice of hippocampus labelled with CTXβ. Right, zoom of retrogradely labelled neurons in boxed region. Scale bars = 500 μm (Left) 100 μm (Right). Image representative of slices from 5 injections. **c** Normalised fluorescence intensity profile of labelled cells in example slice in (**b**). Arrows highlight the two distinct peaks of fluorescence at the two extremes of the radial axis. **d** Two sample hippocampal transverse sections labelled with CTXβ (left, top panels) and the color-coded cells within the sections clustered using a GMM (left, bottom panels). Right, BIC scores for GMM clustering of cells using a range of 1–6 components. Note that for both examples, 2 components produce the lowest score. Scale bar = 100 μm. **e** Summary of BIC scores produced after clustering hippocampal slices from all injections investigated. Note that 2 components consistently showed the lowest BIC score. $n = 5$ injections. **f** Schematic of CTXβ injection into PFC and retrograde labelling in coronal hippocampal slices. **g** Left, Examples of CTXβ labelling in coronal hippocampal slices at AP level −3.18 and −3.58. Right, vH-PFC neurons aligned to the HGEA atlas. Scale bar = 1 mm. **h** Quantification of vH-PFC neurons in CA1 and subiculum. $n = 4$ injections. **i** AAVretro-Switch injection into the PFC of *Calb1-IRES-Cre* mice and subsequent cre-dependent retrograde labelling in hippocampus. **j** Transverse slice of hippocampus labelled with *AAVretro-Switch*. Cyan labels *Calb1*+ PFC-projecting neurons and magenta labels *Calb1*− neurons. Right, zoom of retrogradely labelled neurons in boxed region. Scale bar = 500 μm (Left) 100 μm (Right). Image representative of slices from 4 injections. **k** Normalised fluorescence intensity profile for *Calb1*+ (cyan) and *Calb1*− (black) neurons. Arrows highlight the two genetically distinct peaks of fluorescence at the two extremes of the radial axis. Box plots show median, 75th (box) and 95th (whiskers) percentile.

(Supplementary Fig. 2a, b). Consistent with previous studies we found that for both NAc and LH experiments only a small percentage of neurons were labelled with both tracers, suggesting they sent collaterals to both regions[15,16]. We next looked at how these dual-projecting neurons distributed across the two layers of vH-PFC neurons. Interestingly we found that of all dual-labelled neurons, almost 80% were in the deep layers, with very few in the superficial layer (Supplementary Fig. 2c). Thus, vH neurons that project to PFC have only very limited collateralization, but those neurons with collaterals are preferentially found in the deep, not superficial layer.

We next utilized the expression of calbindin (*Calb1*)—a gene known to be specific to the superficial layer in CA1[7,32], and subiculum[26,33] to target the two vH-PFC layers. We first performed a control analysis where we confirmed the selective expression of calbindin in superficial layers at the CA1/subiculum border in vH (Supplementary Fig. 3a, b), and that this superficial expression overlapped with superficial layer vH-PFC projection neurons labelled with CTXβ. Next, we injected a retrograde AAV expressing a fluorescence switch cassette into the PFC of a *Calb1-IRES-cre* mouse line. In this experiment, the color of the fluorophore expressed depended on the presence of cre (Fig. 1i–k). Neurons projecting to PFC, that also express *Calb1* (and therefore *cre*) will be labelled with one fluorophore, while neurons projecting to PFC that do not express *Calb1* will be labelled with a different fluorophore[34]. The expression of *Calb1* robustly differentiated each population (Fig. 1j, k), where the majority of superficial layer vH-PFC neurons expressed *Calb1*, while the majority of deep layer neurons did not (Supplementary Fig. 3c–e).

Together, these experiments show that there are two vH-PFC projection populations that have distinct molecular properties and are located in the superficial and deep layers of the CA1/subiculum border.

**Superficial and deep vH-PFC populations have distinct cellular and morphological properties consistent with their distribution across the radial axis.** To investigate if the properties of the two vH-PFC layers corresponded to previously identified circuits in superficial and deep layer hippocampus, we carried out targeted whole-cell current-clamp recordings in acute transverse slices of vH from mice previously injected with a retrograde tracer in PFC. We focused our analysis where the layered structure was most apparent in transverse slices (see zoomed area in Fig. 1b) – which our previous analysis suggests is proximal subiculum. However it is important to note that this area has also been frequently described as both as CA1 and subiculum (see Supplementary Fig. 1 and discussion[10,12,16,26,35,36]). By recording sequential pairs of retrogradely labelled neurons in the superficial and deep layers,

this allowed us to compare intrinsic electrophysiological properties. In addition, filling the neurons with a morphological dye during the recording allowed us to reconstruct the dendritic morphology of a subset of these recorded neurons using two-photon microscopy. Superficial layer neurons were morphologically compact, with early branching of the apical dendrite, while deep layer neurons tended to have a relatively sparse, but long apical dendritic tree (Fig. 2b–d, Supplementary Fig. 4). Electrophysiologically, superficial neurons were predominantly regular firing with high $I_h$, while deep layer neurons were more likely to burst fire, and had a more subtle $I_h$ (Fig. 2e–h, Supplementary Fig. 4). These recordings further suggested that the vH-PFC projection is composed of two populations of neurons, each arising from the classically described superficial and deep layers of the CA1/subiculum border[13,14,37,38].

**Superficial and deep vH-PFC neurons are differentially connected to local and long-range input.** vH is innervated by a wide range of afferent input, the identity of which is strongly influenced by both the spatial location and downstream projection of the neuron[13,16,39]. Our results thus far suggest the two populations of vH-PFC neurons may be poised to receive different afferent input dependent on their downstream projection (to PFC), but also dependent on their spatial location along the radial axis (superficial or deep). To investigate if this was the case, we first used tracing the relationship between input and output (TRIO)—a rabies virus based retrograde tracing technique that allowed us to trace the input arriving specifically onto vH neurons projecting to PFC. Using TRIO we found dense input onto vH-PFC neurons from a number of local and long range regions (Supplementary Fig. 5).

We next used channelrhodopsin assisted circuit mapping (CRACM) to investigate the functional input to each layer (Fig. 3). We used AAV injections to express channelrhodopsin (ChR2) using a pan-neuronal synapsin promoter in each of the 4 most densely labelled areas identified in our TRIO experiment – hippocampal CA3, medial entorhinal cortex (ENT), a disperse cluster of anterior thalamic regions focused around paraventricular thalamus (ATh) and an area encompassing the ventral medial septum and the diagonal band of Broca (DBB, see Supplementary Fig. 5). After 2 weeks to allow for virus expression, we made acute transverse slices of vH and performed sequential paired whole cell current-clamp recordings from retrogradely labelled PFC-projecting neurons in each layer. Using brief pulses of blue (473 nm) light allowed us to directly compare the relative influence of synaptic input at the soma of superficial or deep layer vH-PFC neurons from each of the afferent regions[40]. We found that while CA3 terminal stimulation resulted

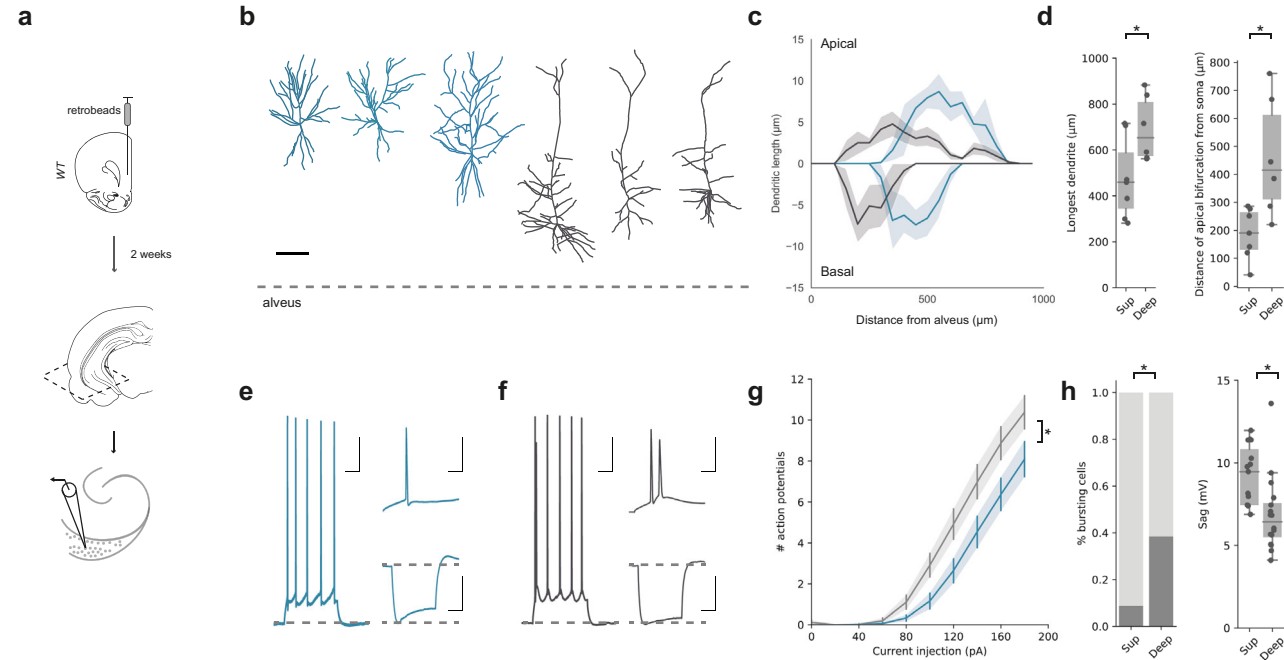

**Fig. 2 Superficial and deep vH-PFC neurons have distinct electrophysiological and morphological properties. a** Schematic of fluorescent bead (retrobeads) injection into PFC and retrograde labelling in transverse hippocampal slices prepared for recordings. **b** Example reconstructions of superficial (blue) and deep (dark grey) PFC-projecting hippocampal neurons. Scale bar = 100 μm, dotted line represents alveus. Images representative of 7 superficial and 6 deep neurons. **c** Sholl analysis showing average dendritic length of apical (top) and basal (bottom) dendrites with increasing distance from the alveus (lateral) for superficial (blue) and deep (grey) layer vH-PFC neurons. Note that each population samples a distinct range of the radial hippocampal axis. (Apical: repeated measures ANOVA, Main effect of group: $F_{(35,385)} = 1.549$, $p = 0.027$; basal: repeated measures ANOVA with Greenhouse-Geisser correction, Main effect of group: $F_{(33,367)} = 1.701$, $p = 0.179$; $n = 7$ (superficial), $n = 6$ (deep) neurons from 6 mice). **d** Left, quantification of the distance from the farthest dendrite tip to the soma (two-sided Mann Whitney, $U = 35.5$, *$p = 0.045$). Right, quantification of the distance of the apical bifurcation from the soma in superficial (Sup) and Deep neurons (two-sided Mann Whitney, $U = 38.5$, *$p = 0.015$, $n = 7$ (superficial), $n = 6$ (deep) neurons from 6 mice). **e** Left, example response of a superficial layer PFC-projecting cell in response to a current injection of 140 pA. Top right, detail of first 50 ms of current injection. Bottom right, response to a current injection of –160 pA. Scale bars = 100 ms, 20 mV; 10 ms, 20 mV; 100 ms, 20 mV. **f** As in (**e**) but for a neighboring deep layer PFC-projecting neuron. Note burst firing in response to current injection, and lower level of voltage sag after negative current injection. **g** Quantification of number of action potentials elicited by somatic current injection in superficial (blue) and deep (black) neurons (Repeated measures ANOVA with Greenhouse-Geisser correction, Interaction between group and current step: $F_{(1.4, 44.5)} = 4.3$, *$p = 0.032$, $n = 22$ (sup) and 26 (deep) from 12 mice). **h** Left, quantification of the proportion of bursting neurons after positive current injections (Fishers Exact test, Odds ratio = 12.5, *$p = 0.009$). Right, quantification of sag amplitude after negative (–160 pA) current injections (two-sided Mann Whitney, $U = 39.5$, *$p = 0.002$, $n = 22$ (sup) and 26 (deep) from 12 mice). Box plots show median, 75th (box) and 95th (whiskers) percentile. Line plots show mean (line) and SEM (shading and error bars).

in roughly equal excitatory synaptic drive in both superficial and deep neurons, the influence of ENT input was biased towards superficial layer neurons, and the influence of both DBB and ATh input was markedly biased towards deep layer neurons (Fig. 3a–d). We subsequently confirmed differences in local input from ipsilateral CA3 and ENT using electrical stimulation of the Schafer collaterals and temporoammonic pathway respectively (Supplementary Fig. 5e, f). Thus, the two populations of vH-PFC neurons are differentially driven by afferent input—both populations receive dense CA3 input, but the influence of cortical input is biased towards superficial cells, while in contrast the influence of thalamic and basal forebrain input is biased towards deep cells.

There is strong local inhibition in hippocampal CA1 and subiculum mediated by interneurons[13,14,41]. In particular, the connectivity of parvalbumin positive (PV+) interneurons in vH is dependent on both spatial location along the radial axis, and downstream projection target[41]. Therefore, we next investigated how the two populations of vH-PFC neurons were connected with the local interneuron network.

To investigate the connectivity of local PV+ interneurons onto each layer, we used AAV injections in a *PV-IRES-Cre* mouse to

express ChR2 in PV+ interneurons in vH, and retrograde tracers to label PFC-projecting neurons (Fig. 3e). After 2 weeks we performed sequential paired whole-cell voltage-clamp recordings from each layer, using a high chloride internal to allow for isolation of inhibitory currents. Similar to before, we could then use brief blue light pulses to investigate PV mediated inhibition onto superficial and deep vH-PFC neurons. We found that ChR2-mediated activation of PV+ interneurons in vH resulted in robust, yet equivalent IPSCs in both superficial and deep layer vH-PFC neurons (Fig. 3e). This was surprising, as it is in contrast to previous reports of preferential inhibition of deep layer neurons by PV+ interneurons[41]. Therefore, in the same slices we recorded neighboring unlabelled superficial and deep neurons, and confirmed that non-PFC-projecting vH neurons in the deep layers receive more PV+ input than superficial layers (Fig. 3f). Thus, vH-PFC projecting neurons are specifically connected to ensure equivalent inhibition across the two layers.

Next, to investigate the excitatory input onto local interneurons arising from vH-PFC neurons in each layer, we combined an injection of retrograde AAV expressing cre recombinase into PFC, and an injection into vH of a mixture of AAV to express cre-dependent fluorescently tagged somatic channelrhodopsin

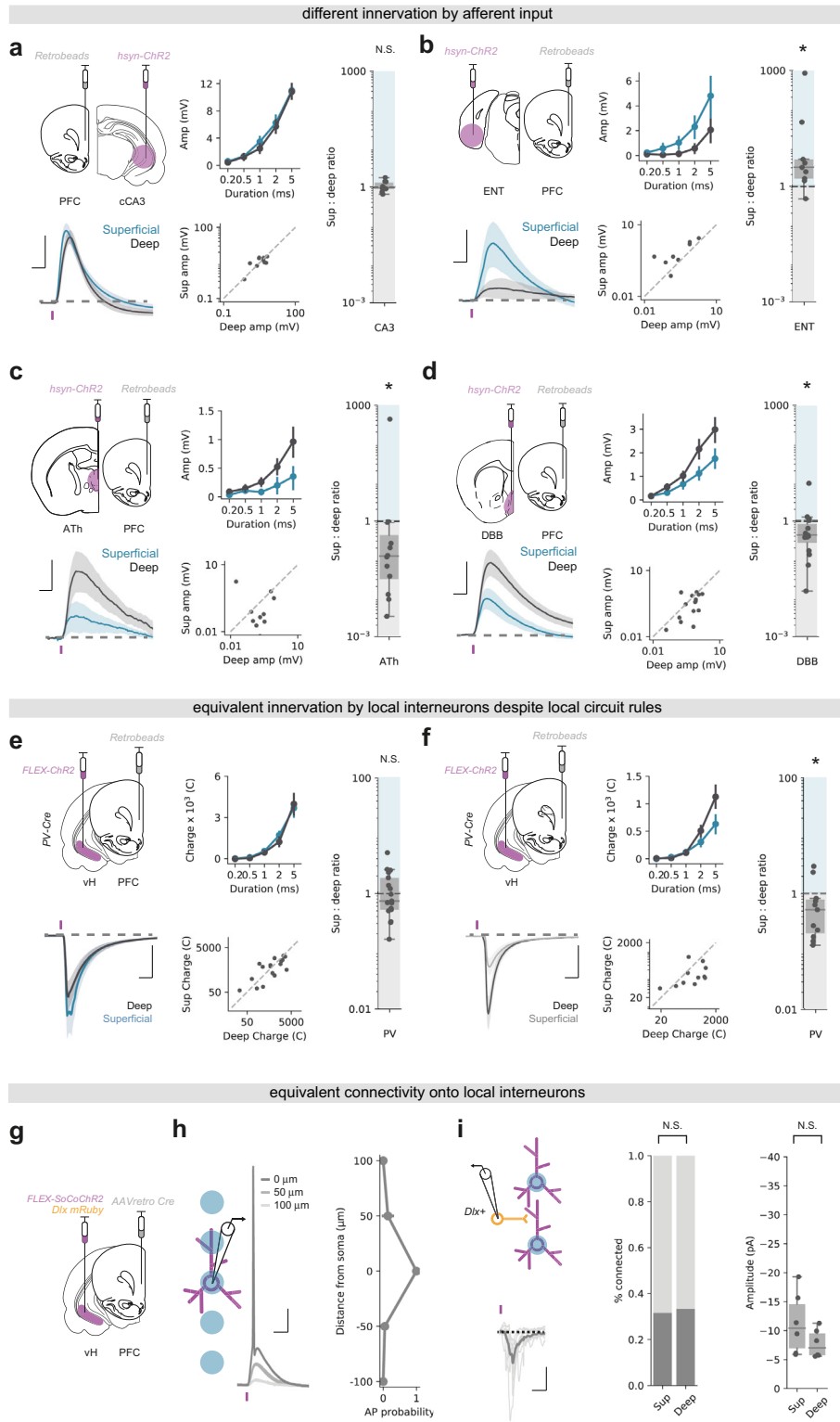

(soCoChR2)[42], and the fluorescent marker mRuby under the control of the interneuron specific *dlx* promoter[43,44]. We first confirmed that the *dlx* promoter was a suitable tool to identify interneurons in vH (Supplementary Fig. 6). Next, we ensured that soCoChR2 expression allowed for light induced action potential firing with an appropriate spatial resolution. We recorded from soCoChR2 expressing neurons in the presence of synaptic

blockers, and used a focused light spot to activate soCoChR2 in 50 μm increments centred around the soma of the recorded neuron. Consistent with previous reports we found that the ability to fire action potentials was limited to within 50 μm (Supplementary Fig. 7). In a separate experiment we also confirmed that soCoChR2 negative cells were not light sensitive. This approach therefore allowed us to elicit action potentials in

**Fig. 3 Superficial and deep vH-PFC neurons are differentially connected to local and long-range input. a** Top left, Schematic showing experimental setup. ChR2 was injected into contralateral CA3 and retrobeads injected into PFC. 2 weeks later input-specific connectivity was assessed using paired recordings of superficial and deep vH-PFC neurons in acute slices. Bottom left, Average light-evoked responses in pairs of superficial (blue) and deep (black) layer PFC-projecting hippocampal neurons in response to cCA3 input. Scale bar = 10 ms, 5 mV. Purple tick represents the light stimulus. Middle, summary of amplitude of Sup and Deep responses to increasing durations of light pulse (top), and amplitudes of individual pairs at 5 ms (bottom). Right, summary of the ratio of superficial: deep neuron EPSP amplitude. Higher values mean input is biased to superficial neurons, low values towards deep layer neurons. Note log scale. CA3 input is equivalent onto superficial and deep layer neurons (two- sided Wilcoxon paired test, $W = 27$, N.S.$p = 1.0$, $n = 10$ pairs of neurons from 3 mice). As in (**a**) but for ENT (**b**), ATh (**c**) and DBB (**d**) input. Scale bar = 10 ms, 2 mV (**b**), 0.5 mV (**c**), 1 mV (**d**). ENT input is biased towards superficial layer neurons (two- sided Wilcoxon paired test, $W = 44$, *$p = 0.007$, $n = 9$ pairs of neurons from 3 mice), while both ATh and DBB are biased towards deep layer neurons (ATh: two- sided Wilcoxon paired test, $W = 12$, *$p = 0.034$, $n = 7$ pairs of neurons from 4 mice; DBB: two- sided Wilcoxon paired test, $W = 24$, *$p = 0.041$, $n = 15$ pairs of neurons from 7 mice). **e** As in (**a**) but for local PV interneuron input. Scale bar = 10 ms, 500 pA. PV + inhibitory input is equivalent onto superficial and deep layer PFC-projecting neurons (two-sided Wilcoxon paired test, $W = 57$, *$p = 0.38$, $n = 17$ pairs of neurons from 4 mice). **f** As in (**e**) but in neighboring, unlabelled vH neurons from superficial and deep layers. Scale bar = 10 ms, 200 pA. PV + inhibitory input is biased towards non-retrogradely labelled deep layer neurons (two-sided Wilcoxon paired test, $W = 10$, *$p = 0.042$, $n = 11$ pairs of neurons from 3 mice). **g** Strategy to investigate superficial and deep vH neuron connectivity onto local interneurons. **h** Focused light allows activation of neurons expressing soCoChR2 with high spatial resolution. Scale bar = 10 mV, 20 ms. **i** Connectivity of superficial and deep PFC-projecting vH neurons onto neighboring $dlx+$ interneurons. Probability of connectivity and amplitude is equivalent for both layers (connectivity: Fishers Exact test, Odds ratio = 0.95, N.S.$p = 1.0$, $n = 19$ (sup) and 18 (deep) neurons from 4 mice; amplitude: two-sided Mann Whitney, $U = 8$, N.S.$p = 0.42$, $n = 5$ (sup and deep) connected neurons from 4 mice). Scale bar = 10 pA, 20 ms. Box plots show median, 75th (box) and 95th (whiskers) percentile. Line plots show mean (line) and SEM (shading and error bars).

individual PFC-projecting neurons from each layer using a focused light spot to activate somatically targeted CoChR2, while recording from neighboring genetically identified interneurons in vH (Fig. 3g, h). Using this approach, we again found that PFC-projecting neurons in each layer connected to local interneurons with a similar connection probability and synaptic strength (Fig. 3i). Combined with the disparate afferent input identified above, this suggests that the two vH-PFC circuits are connected in a way that might facilitate lateral inhibition across the two layers, to promote their activity at distinct timepoints.

**Superficial and deep vH-PFC neurons connect differentially in PFC.** Our results so far suggest the presence of two populations of vH-PFC neurons that are differentially controlled by local and long-range input. We next asked if these populations might connect differentially in PFC.

We investigated whether superficial and deep layer vH-PFC neurons innervated unique spatial locations within PFC. To do this we injected an AAV into the vH of *Calb1-IRES-cre* mice to express either ChR2 where expression was restricted to only *cre*-expressing (superficial layer) neurons (creON ChR2). In a separate experiment we injected an AAV where ChR2 was inhibited in *cre*-expressing neurons[45], and thus was only expressed in *cre*-negative (deep layer) neurons (creOFF ChR2, Supplementary Fig. 8a). We then made coronal slices of PFC from these animals to investigate the distribution of axons from these neurons. We found that there were no obvious differences in the distribution of axons from the two layers across the AP, ML or DV axes (Fig. 4a, b). This suggests that the two layers of vH-PFC neurons, despite receiving distinct afferent input, innervate similar areas and layers of PFC.

We next asked whether the two layers made unique connections onto excitatory and inhibitory neurons in PFC. We first used monosynaptic rabies tracing from either inhibitory interneurons (using $VGAT+$ starter cells) or excitatory neurons (using $CaMKii+$ starter cells) in PFC (Fig. 4d). We compared neurons retrogradely labelled with rabies in vH, with the distribution of neurons labelled with a simultaneous injection of CTXβ in PFC. Therefore, in this experiment all vH-PFC neurons are labelled with CTXβ, while either vH-PFC^excitatory or vH-PFC^inhibitory neurons are labelled with rabies. We then split labelled neurons into two layers using the unbiased GMM approach outlined in Fig. 1. Following GMM analysis, each rabies

labelled neuron was classified as being in either the superficial or deep cluster (Fig. 4e) and the proportion of rabies labelled neurons in each vH layer for either *VGAT* or *CaMKii* starter cells in PFC could be calculated. Using this strategy, we found differences in the targeting of excitatory and inhibitory neurons in PFC by each layer in vH (Fig. 4f). While, both superficial and deep layer vH neurons targeted interneurons in PFC, input to pyramidal neurons in PFC preferentially originated from neurons in deep layers.

Our tracing experiment suggested that the superficial layers connect more readily with inhibitory interneurons in PFC, while deep layer neurons connect with a mixture of both inhibition and excitation. To confirm this distinct targeting using CRACM, we used *Calb1-IRES-cre* mice, and injected an AAV to express either creON ChR2, or creOFF ChR2 as before (Fig. 4g). We first confirmed that there was no contamination of *Calb1* creON innervation from putative long-range inhibitory projections that may also express calbindin by investigating if any vH-PFC neurons were positive for the inhibitory marker *VGAT* (Supplementary Fig. 8b, c). We then carried out whole-cell voltage-clamp recordings of deep layer pyramidal neurons in acute slices of PFC from these animals in the presence of the NMDA-receptor antagonist APV, and assessed the relative excitatory and inhibitory drive. We did this by recording light-evoked synaptic input at –70 mV (predominantly AMPA-mediated excitatory currents) and 0 mV (predominantly GABA-mediated inhibitory currents, mediated by disynaptic feedforward inhibition from local interneurons Fig. 4g). We chose to record from deep, layer V neurons in PFC as our anatomy and that from previous studies[46] have shown that the majority of input from vH arrives into these layers (although see discussion). By comparing the ratio of responses at –70 mV and 0 mV, we confirmed that the superficial layer of vH drives substantially more feedforward inhibition in PFC compared to the deep layer (Fig. 4i).

Interneurons in PFC can be characterized into two main subgroups—soma-targeting fast-spiking interneurons, and dendrite-targeting non-fast-spiking interneurons—based on their intrinsic properties and the expression of peptides such as parvalbumin and somatostatin[47]. As both deep and superficial layer vH neurons could drive feedforward inhibition—albeit to different extents—we next wanted to see if the inhibitory circuitry each layer contacted was different. To do this we again used CRACM to compare the relative input from superficial and deep layer vH neurons into PFC. However, we

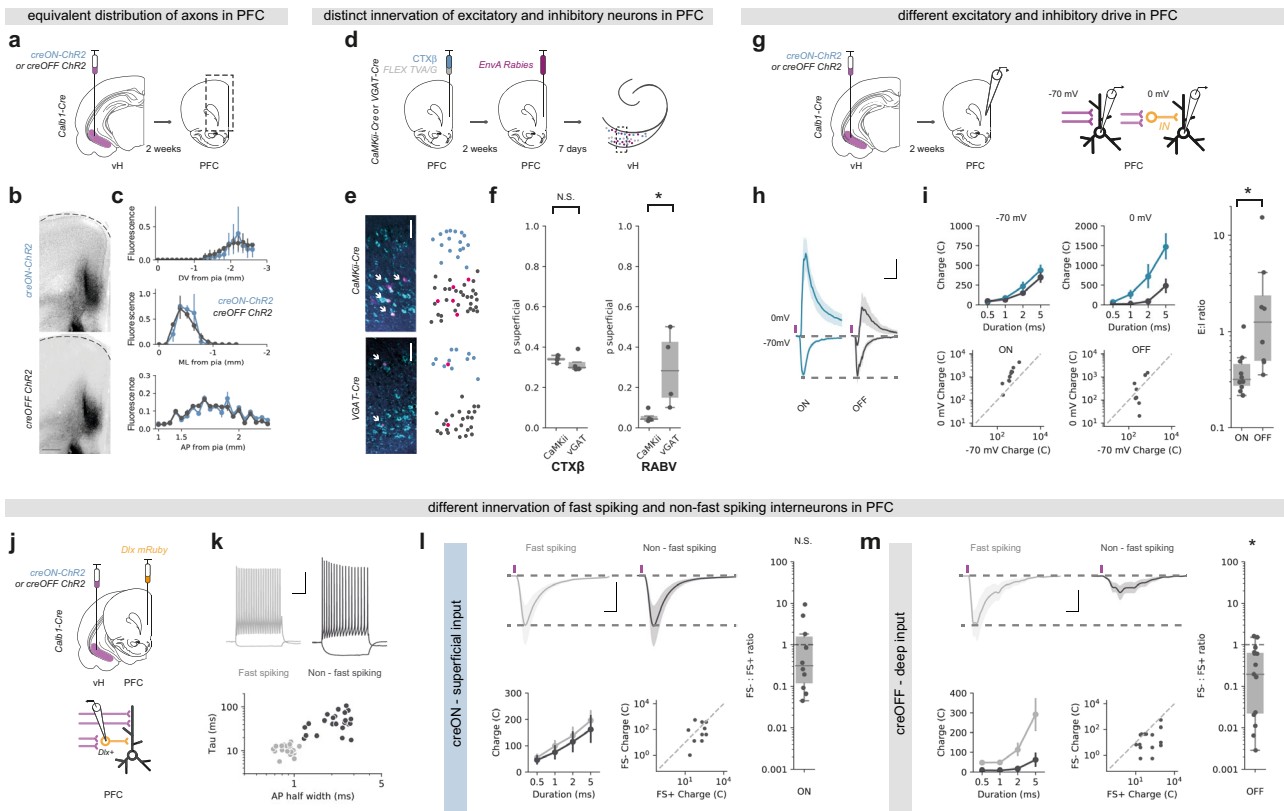

**Fig. 4 Superficial and deep vH-PFC neurons connect differentially in PFC. a** Strategy to label axons from superficial (creON) and deep (creOFF) layers for visualization in PFC. **b** Example images of superficial (top) and deep (bottom) layer vH axon terminals in PFC. Scale bar = 100 μm. Images representative of slices from 4 injections. **c** Distribution of superficial (creON, blue) and deep (creOFF, grey) layer axon terminals in PFC along the DV, ML and AP brain axes. Note the distribution for both layers is equivalent (repeated measures ANOVA, main effect of group, DV: $F_{(1,6)} = 0.31$, $p = 0.599$; ML: $F_{(1,6)} = 0.10$, $p = 0.77$; AP: $F_{(1,6)} = 0.03$, $p = 0.59$; $n = 4$ injections). See supplementary statistics table for more information. **d** Strategy to label neurons projecting to inhibitory and excitatory neurons in PFC. **e** Left, transverse slice of hippocampus labelled with CTXβ (cyan) and rabies (magenta) after tracing from excitatory (*CaMKii*, top) or inhibitory (*VGAT*, bottom) neurons in PFC. Note the restriction of rabies labelling from excitatory neurons to the deep layer. Scale bars = 100 μm. *Right*, product of GMM clustering (see Fig. 1 and "Methods") of all cells in the example image. Superficial layer is shown in blue, deep layer in black and rabies labelled cells superimposed in magenta. **f** Proportion of CTXβ positive neurons (left) and rabies positive neurons (right) in the superficial layer for both experiments (*CaMKii*, vH^excitatory and *VGAT*, vH^inhibitory). Note equivalent distribution of CTXβ across both conditions (two-sided Mann Whitney, $U = 4$, ^N.S.$p = 0.3$), but a marked absence of neurons projecting to excitatory PFC neurons (RABV: *CaMKii*) in the superficial layer (two-sided Mann Whitney, $U = 16$, *$p = 0.03$, $n = 4$ injections). **g** Strategy to record PFC excitatory input, or inhibitory input mediated via feedforward interneurons (orange). **h** Responses to superficial (blue) or deep (grey) hippocampal inputs at -70 mV (EPSCs) and 0 mV (IPSCs) in deep layer PFC neurons. Purple tick indicates light pulse. Scale bar = 20 ms and 0.5 (fold response amplitude at –70 mV, which is normalized to 1). **i** *Left*, summary of amplitude of superficial (creON, blue) and deep (creOFF, grey) responses at –70 mV and 0 mV to increasing durations of light pulse (top), and recorded amplitudes of individual responses at –70 mV and 0 mV to superficial (ON) and deep (OFF) input at 5 ms (bottom). Right, summary of the ratio of responses recorded at –70 mV: 0 mV. Higher values mean input is biased to excitation, low values towards inhibition. Note log scale. Input from superficial neurons has a greater inhibitory contribution than that of deep neurons (two sided Mann Whitney, $U = 10$, $p = 0.006$, $n = 10$ neurons 4 from mice (sup), 8 neurons from 3 mice (deep)). **j** Strategy to record input from the two layers of vH onto identified interneurons in PFC. **k** *Top*, example current clamp recordings from fast-spiking (FS +, grey) and non-fast-spiking (FS-, black) interneurons in PFC. Bottom, summary showing clustering of recorded neurons into two groups based on membrane time constant and action potential half width. **l** Top, responses to superficial (creON) input at –70 mV (EPSCs) onto neighboring fast spiking (grey) or non-fast spiking interneurons (black). Scale bar = 10 ms, 200 pA. Bottom, summary of amplitude of FS + and FS-responses to increasing durations of light pulse (left), and amplitudes of individual pairs at 5 ms (*right*). Right, summary of the ratio of FS-: FS+. Higher values mean input is biased to non-fast spiking (FS-), low values towards fast spiking (FS+) neurons. Note log scale. Superficial input is equivalent onto FS + and FS- neurons in PFC (two-sided Wilcoxon paired test, $W = 35$, ^N.S.$p = 0.49$, $n = 10$ pairs from 5 mice). **m** As in (**l**) but for deep (creOFF) input. Deep input is biased towards fast spiking (FS +) interneurons in PFC (two-sided Wilcoxon paired test, $W = 111$, *$p = 0.02$, $n = 15$ pairs from 4 mice). Box plots show median, 75th (box) and 95th (whiskers) percentile. Line plots show mean (line) and SEM (shading and error bars).

used an injection of AAV to express *dlx* mRuby into PFC to allow targeted whole cell recordings from inhibitory interneurons (Fig. 4j). Using this approach, we classified each interneuron as either fast spiking or non-fast spiking based on intrinsic electrophysiological properties (Fig. 4k), and then examined the relative input onto neighboring pairs of neurons of each type from either the superficial or deep layer of vH. Superficial vH input innervated both fast-spiking and non-fast-spiking interneurons in PFC to an equal extent, suggesting widespread recruitment of both dendritic and somatic inhibitory circuits within PFC (Fig. 4l). In contrast, deep layer vH input was very selective—while it showed reliable input onto fast-spiking interneurons in PFC, there was little to no input onto neighboring non-fast-spiking interneurons (Fig. 4m). Together this suggests that vH-PFC neurons are connected in such a way to have differential excitatory and inhibitory influence on PFC.

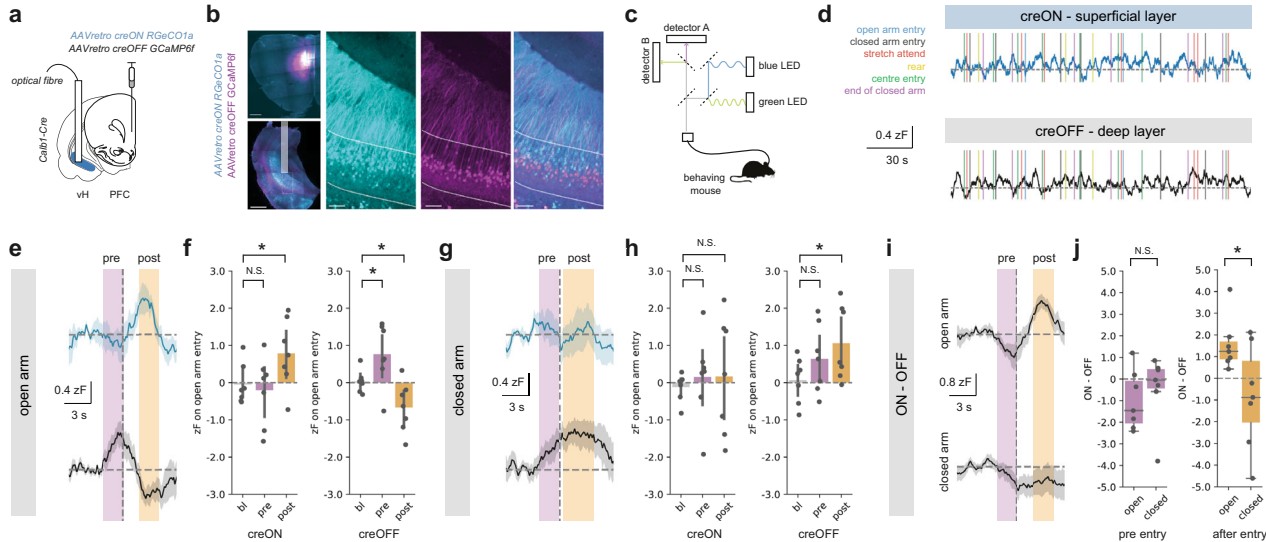

**Fig. 5 Superficial and deep vH-PFC neuron activity differs during arm entry in the EPM. a** Strategy to record superficial and deep vH-PFC neuron calcium dynamics during free behaviour. **b** Left, example images of the viral strategy implemented for imaging. Left, injection of retrograde calcium sensors in PFC (top) and photometry fibre implant in vH (bottom). Scale bars = 500 μm. Right, zoom of creON RGeCO1a (cyan) and creOFF GCaMP6f (magenta) calcium sensor expression in the superficial and deep layers of vH, respectively. Scale bars = 100 μm. Images representative of slices from 7 mice. **c** Light path and recording setup for dual signal detection during free behaviour, allowing simultaneous detection of superficial and deep layer activity, see also Supplementary Fig.9. **d** Example traces of creON (superficial layer) and creOFF (deep layer) calcium signal while a mouse is behaving during EPM exploration. **e** Summary traces of superficial (top, blue) and deep (*bottom*, grey) layer activity aligned to open arm entry (dashed line) during EPM exploration. Activity periods pre (magenta) and post (orange) arm entry are highlighted for subsequent quantification. **f** Summary of peri-event normalized calcium-dependent fluorescence recorded from superficial layer (creON) and deep layer (creOFF) cells during the baseline (bl; −6 s to −5 s before arm entry), pre and post arm entry periods for open arm entries. Superficial (creON) neurons increase activity on open arm entry (Repeated measures ANOVA, Main effect of epoch: $F_{(2,12)} = 4.52$, *$p = 0.034$; planned two-sided Welch's paired t-test with Holm-Šidák correction: bl vs. pre: $t_{(6)} = 0.3$, N.S.$p = 0.77$; bl vs post: $t_{(6)} = 4.4$, *$p = 0.004$, $n = 7$ mice), while deep (creON) neurons increase pre-entry and decrease activity post-entry (Repeated measures ANOVA, Main effect of epoch: $F_{(2,12)} = 13.4$, *$p = 0.001$; planned two-sided Welch's paired t test with Holm-Šidák correction: bl vs. pre: $t_{(6)} = 2.5$, *$p = 0.04$; bl vs post: $t_{(6)} = 3.4$, $p = 0.02$, $n = 7$ mice). See supplementary statistics table for further information. **g** As in (**e**) but for closed arm entry. **h** As in (**f**) but for closed arm entry. In contrast to open arm entries, superficial layer activity remains unchanged (Repeated measures ANOVA, Main effect of epoch: $F_{(2,12)} = 0.11$, N.S.$p = 0.90$; planned two-sided Welch's paired t test with Holm-Šidák correction: bl vs. pre: $t_{(6)} = 0.8$, N.S.$p = 0.43$; bl vs post: $t_{(6)} = 0.06$, N.S.$p = 0.95$, $n = 7$ mice), while deep layer neurons show a sustained increase in activity around and upon closed arm entry. **i** As in (**e**) and (**g**) but for the difference between simultaneously recorded superficial (ON) and deep (OFF) layer activity. **j** Summary showing the difference between superficial (ON) and deep (OFF) activity pre- (magenta, left) and post- (orange, right) entry into the open and closed arms. Values above 0 indicate a bias towards superficial layer activity and values below 0 indicate a bias towards deep layer activity. Note that while activity pre-entry into both open and closed arms does not show a layer bias (two-sided Welch's paired t test with Holm-Šidák correction, $t_{(6)} = 0.6$, N.S.$p = 0.59$), activity post-entry is biased towards the superficial layer for open arm and towards deep layer cells for closed arm entries (two-sided Welch's paired t test with Holm-Šidák correction, $t_{(6)} = 3.1$, *$p = 0.02$, $n = 7$ mice). Box plots show median, 75th (box) and 95th (whiskers) percentile. Line plots show mean (line) and SEM (shading and error bars), bar charts show mean (bar) and SEM (error).

**Superficial and deep vH-PFC neurons have opposing activity around open and closed arm entry in the EPM**. The two populations of vH-PFC neurons are connected in a way that may support opposing activity. Thus, we next asked if the superficial and deep populations in vH were differentially active during approach avoidance behaviour.

To do this, we simultaneously recorded the activity of both populations using bulk calcium imaging through an optical fibre (Fig. 5a–d). We used an injection of a mixture of retrograde AAVs into PFC to express both creON RGeCO1a – a red wavelength calcium indicator, and creOFF GCaMP6f—a green wavelength calcium indicator - in *Calb1-IRES-cre* mice. Using this technique, *Calb1 +*, superficial neurons in vH projecting to PFC will express RGeCO1a, and *Calb1-* deep neurons projecting to PFC will express GCaMP6f. By collecting green and red fluorescence through an implanted optical fibre (Supplementary Fig. 9), this allowed simultaneous monitoring of both superficial and deep layer vH-PFC neuron activity in freely behaving mice while they explore the EPM.

We aligned these recordings to the onset of different behaviours exhibited by mice as they explored the maze. Interestingly, activity surrounding rearing, stretch attend and head dipping behaviour—classically seen as investigative behaviour during maze exploration—was not consistently modulated (Supplementary Fig. 10). However, we noticed consistent changes in activity in both layers when mice moved into either the closed arms or the open arm of the EPM, and that this activity was apparent both before entry into the arm, and after arm entry (Fig. 5e–h). Interestingly this activity was very different across the two layers of vH-PFC neurons. We found that upon exploration of the open arms of the maze, superficial neurons remained at baseline prior to arm entry, but transiently increased activity in the period after entry into the arm. In contrast, activity of deep layer neurons increased before entry to the open arm, but this activity was reduced dramatically upon arm entry (Fig. 5e, f). Thus upon entry to the open arm, the superficial and deep layers undergo opposing changes in activity, consistent with our circuit mapping data.

Interestingly these changes in activity were also distinct when mice entered the closed arms of the maze. Superficial layer neurons exhibited no change in activity either before or after entry of the closed arm. Similar to open arm entry, deep layer neurons again increased activity prior to closed arm entry, but in contrast, this activity remained high when the mouse entered the arm (Fig. 5g, h). These changes in activity were transient, and activity returned to baseline levels by the time the animal had reached the end of arm exploration and turned to return (Supplementary Fig. 11a). In addition, changes in superficial layer activity were much more prominent on the first entry to the open arm, and reduced over successive entries. In contrast, deep layer activity remained relatively stable across the session (Supplementary Fig. 11b, c). These results suggest that the difference between the activity of the two layers may distinguish closed and open arm entry. Indeed, on open arm entry, the difference in activity between the two layers was biased to the superficial layer, while in contrast, on closed arm entry activity was biased to the deep layer (Fig. 5i, j). This suggests that the relative activity of the superficial and deep layers of the vH-PFC projection are tightly and oppositely regulated around the choice point of the EPM.

**Optogenetic activation of superficial or deep input into PFC has opposing influence on behaviour in the EPM.** Finally, we wanted to test the causal role of the two vH-PFC populations in the exploration of the EPM. Due to the large differences in connectivity and utilization during behaviour, we hypothesized that artificial activation of the superficial and deep layers may drive opposite behaviour.

We again used an AAV based approach in *Calb1-IRES-cre* mice to express ChR2, or control GFP, in either superficial (creON) or deep (creOFF) layer neurons in vH, and implanted optical fibres unilaterally in PFC (Fig. 6a, b). This allowed us to stimulate axons arriving in PFC from each of the two layers of vH with blue light while the animal was exploring the maze. Consistent with the transient nature of our photometry data, multiple previous studies have shown that transient manipulation of the vH-PFC circuit only around the centre point and open arms is crucial for defining behaviour[4,48]. Therefore, we investigated whether artificially activating either the superficial or deep vH-PFC populations in this manner would influence mouse behaviour on the maze. After a 3 min baseline, we artificially activated axons with blue light for a three-minute epoch, only when mice entered the central choice point in the EPM, and maintained excitation until the mice returned to the closed arms (Fig. 6c). Thus, the activity of each population was only transiently artificially activated based on the mouses movement. We found that this manipulation consistently increased open arm exploration in creON (superficial layer) mice, while in contrast reduced open arm exploration in creOFF (deep layer) mice, compared to GFP controls (Fig. 6d–f). In addition, this manipulation altered the probability of entering the open arms in a similar fashion, where creON mice had a higher probability of entering the open arms during the light epoch, and creOFF mice had a lower probability, when compared to GFP controls (Supplementary Fig. 12).

We next asked whether this behavioural effect could also be seen with constant light stimulation during the 3 min light epoch —stimulation that was not dependent on the position of the mouse. Interestingly, and consistent with previous studies[4,48], this manipulation had minimal effect on behaviour, suggesting that the transient activation of each pathway around the choice point in the EPM is necessary for changes in behaviour (Supplementary Fig. 13). Together these results suggest that activation of superficial layer vH axons in PFC promotes the exploration of

the open arms of the EPM, while activating the deep layer vH axons in PFC reduces exploration of the open arms.

## Discussion
Here we show that the projection from vH to PFC can be subdivided into two populations that are segregated along the radial axis of vH (Figs. 1–3). These parallel populations of pyramidal neurons have opposing influence on both PFC circuit activity (Fig. 4) and behaviour during innate approach avoidance conflict (Figs. 5, 6). The superficially located population is preferentially connected to widespread inhibitory circuitry in PFC, is driven by cortical input and promotes exploration. In contrast, the deep population is preferentially connected to pyramidal neurons and fast spiking interneurons in PFC, is driven by basal forebrain and thalamic input, and promotes avoidance.

**Distribution of vH-PFC neurons across CA1 and subiculum.** We found that vH-PFC neurons are present at the border between CA1 and subiculum (Fig. 1). Interestingly, these neurons were spread in a continuous gradient across the AP axis in almost equal proportions in both CA1 and subiculum. Importantly the border we utilized in or study was defined by the Allen Brain Atlas coordinate space, and the location of this border is very variable across both atlases and studies. For example, vH-PFC neurons are often classified as being located either in vCA1, or as predominantly in subiculum[12,16,17,31] (see Supplementary Fig. 1). In particular it is becoming increasingly evident that in contrast to dorsal hippocampus, which is arranged in a tight, spatial distribution with sharp borders between subfields[17,28,49,50], ventral hippocampal subfields seem to be organized as a gradient, with few distinct anatomical delineations between subfields[16,26,35]. Therefore, in our study we have treated vH-PFC neurons as a continuum that is consistently segregated into two populations across the radial axis. However, it is important to note that the difficulty in defining the CA1 and subiculum border in ventral hippocampus means that our study did not explicitly differentiate these two distinct subregions of vH. It is known that these two subfields perform distinct roles in the hippocampal calculation[49–53], and have distinct local and long-range connectivity[15–17,31]. Thus, our data cannot differentiate this important information. Future work – potentially by using multiple levels of intersectional targeting – is needed to more accurately define the borders between subfields in vH, and to tease apart the differential contribution of superficial and deep CA1-PFC and subiculum-PFC connections.

Despite being organized as a continuum across the CA1 subiculum border, we found that vH-PFC neurons were consistently segregated across the radial axis (Figs. 1, 2). The properties, connectivity and behavioural role of neurons in both CA1 and subiculum are distinct across superficial and deep layers[14,32,35,54–56]. However, there is again debate surrounding the exact organization of these layers. In subiculum, neurons have been described as forming between two and 5 layers[26,29,38]. The recently published Hippocampal Gene Expression Atlas[29] proposed three pyramidal layers in ventral subiculum—layers II, III and IV. When we aligned vH-PFC neurons to this atlas (Fig. 1, Supplementary Fig. 1) we found that superficial vH-PFC neurons aligned well with layer II, however, deep layer neurons were found in both layer III and layer IV, and most often on the border between the two layers. The distribution of vH-PFC neurons was better described by an alternative 5-layered distribution[26] (Supplementary Fig. 1b), where vH-PFC neurons aligned well with layer II, and layer IV, and avoided layers III and V associated with expression of the gene *pcp4*. Together, these results suggest that vH-PFC neurons in CA1 and subiculum are

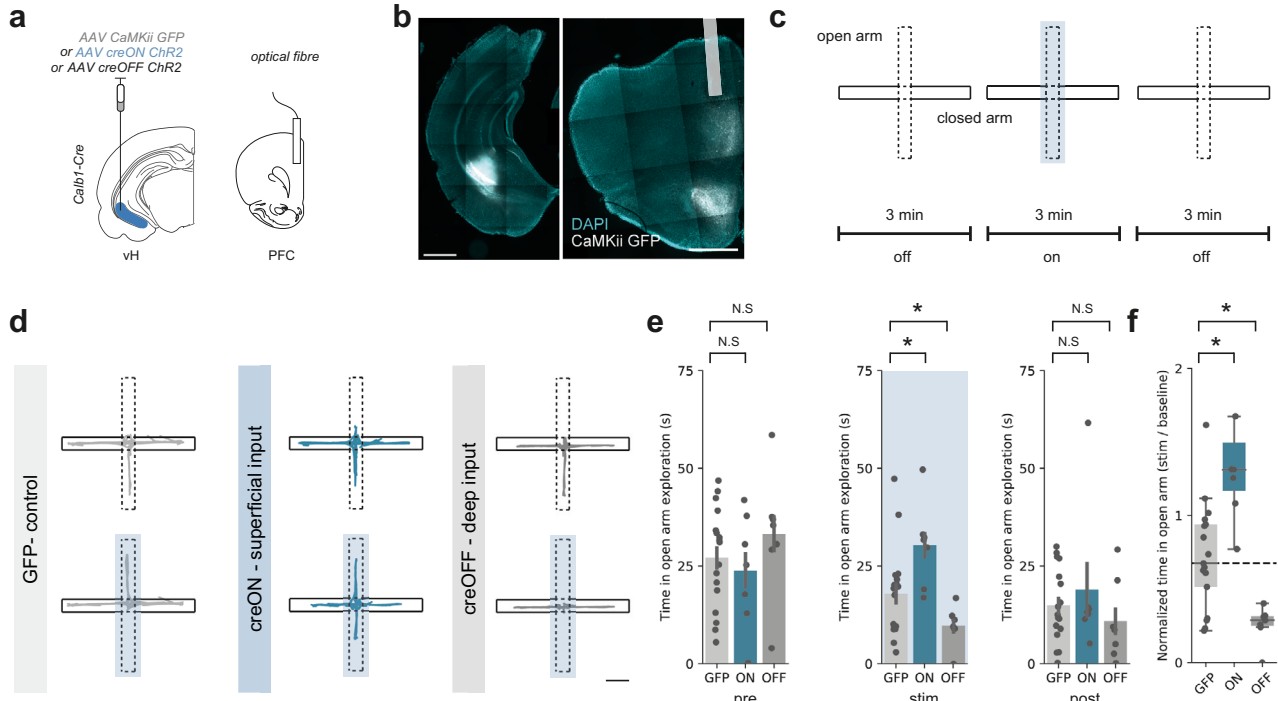

**Fig. 6 Superficial and deep vH-PFC populations bidirectionally influence behaviour in the EPM. a** Strategy for in vivo optogenetic manipulation of vH axons from each layer in PFC. **b** Example images of viral injection into vH (left) and optic fibre implant in PFC (right). Scale bars = 1000 μm. Image representative of slices from all mice used in optogenetic experiments below. **c** Schematic showing experimental design. **d** Trajectories of an example mouse during baseline (top) and during stimulation (bottom) for GFP (left), creON (centre) and creOFF (right) experiments. Scale bar = 10 cm. **e** Summary showing time spent in open arm exploration during the first 3 min epoch of no stimulation (left), the central 3 min with light stimulation (blue box, middle) and the final epoch of no light stimulation (right). Note that differences in time spent in open arm exploration are only evident during the central epoch coupled with light stimulation (Mixed ANOVA, interaction between epoch and group: $F_{(4,56)} = 4.78$, $p = 0.002$; planned two-sided Welch's paired $t$ test: Pre: GFP vs creON, $t_{(10)} = 0.5$, N.S.$p = 0.61$; GFP vs creOFF, $t_{(9.3)} = 0.4$, N.S.$p = 0.39$; Stim: GFP vs creON, $t_{(11.9)} = 2.5$, *$p = 0.03$; GFP vs creOFF, $t_{(21.7)} = 2.4$, *$p = 0.04$; Post: GFP vs creON, $t_{(7.3)} = 0.5$, N.S.$p = 0.61$; GFP vs creOFF, $t_{(10.2)} = 0.9$, N.S.$p = 0.40$, $n = 17$ (GFP), 7 (creON), 7 (creOFF) mice). See supplementary statistics table for further information. **f** Summary of the effect of light-evoked activation on open arm exploration. Superficial layer (creON) stimulation increased, while deep layer (creOFF) decreased exploration relative to controls (GFP) (two-sided Mann Whitney, GFP vs creON: $U = 12$, *$p = 0.002$; GFP vs creOFF: $U = 100$, *$p = 0.008$, $n = 17$ (GFP), 7 (creON), 7 (creOFF) mice). Dotted line shows median exploration of GFP controls for comparison. Box plots show median, 75th (box) and 95th (whiskers) percentile. Bar charts show mean (bar) and SEM (error).

distributed across the CA1 subiculum border, but that this distribution is segregated across the radial axis. This segregation corresponds to classical superficial and deep layers of CA1, but also to the previously defined layered structure of the subiculum.

**Afferent input into vH-PFC neurons is consistent with distinct roles for each layer**. We used a combination of whole brain anatomy and CRACM to investigate the connectivity of these two populations of neurons. Superficial vH-PFC neurons receive strong input from medial entorhinal cortex[32,35] (Fig. 3b), in contrast to superficial neurons in dorsal hippocampus, which receive preferential input from lateral entorhinal cortex[32]. This is consistent with medial entorhinal cortex providing the predominant input to the ventral location occupied by vH-PFC neurons[57]. Medial entorhinal cortex is classically associated with encoding structured representations of the environment, and acts as a relay to return information from PFC to the hippocampus[57,58]. As vH-PFC neurons to not receive input from the other relays from PFC such as nucleus reuneins[16] or amygdala[16,59] (Supplementary Fig. 5), this suggests superficial neurons may preferentially receive structured feedback from prefrontal cortex via medial entorhinal cortex. This is consistent with recordings in PFC that show that structured representation of the arms of the EPM in PFC is critical for open arm exploration[60]. Thus, the high activity of superficial vH-PFC

neurons during open arm entry observed in our study (Fig. 5) may represent the activation of this feedback circuit and thus bias behaviour towards exploration and approach—consistent with the artificial activation of this layer—Fig. 6.

In contrast, deep layer vH-PFC neurons receive preferential input from subcortical areas such as anterior thalamus and the diagonal band (Fig. 3c, d). These areas are classically associated with stress, value, salience and attention[14,39]. This suggests that deep vH-PFC neurons may have a preferential role in behaviour utilizing such affective information, which is in keeping with its role in avoidance during EPM exploration (Figs. 5, 6). Interestingly, we also found that the large majority of neurons that send collaterals to other areas important for salience, affect and avoidance such as NAc and LH[4,31] also arise preferentially from the deep layer of vH-PFC neurons (Supplementary Fig. 2). Finally, it is also interesting to note that Ant Thal, DBB and deep layer vH neurons are all strongly theta modulated when compared to superficial layer neurons[14,39], and neurons in PFC that preferentially encode anxiogenic environments are strongly coupled to hippocampal theta[18,48,60].

**Connectivity of vH neurons in in PFC is consistent with opposing roles during behaviour**. We found that superficial vH-PFC neurons preferentially connect with wide feedforward inhibition in PFC, including strong input onto dendrite-targeting

non-fast spiking interneurons (Fig. 4). This is consistent with a role in promoting exploration and approach behaviour in the EPM (Figs. 5, 6), as dendrite targeting interneurons have been implicated in ongoing utilization of spatial memory, and the promotion of approach and exploration[21,61]. In contrast, deep layer vH input preferentially connects directly with pyramidal neurons and with fast-spiking interneurons, which have been implicated in the promotion of avoidance[19,20,62], again consistent with the effect of deep layer vH-PFC activation in the EPM (Figs. 5, 6). Together this suggests a key role for hippocampal input from each layer in dynamically controlling the utilization of excitatory and inhibitory PFC circuitry to control approach and avoidance behaviour.

It is interesting to note however that there may be differences in the role of vH input across layers of PFC. In this study we investigated the effect of vH input into the deep layers of PFC, which coincides with the highest density of axonal innervation[46,63] (Fig. 4a, b). However, there is also substantial vH input into more superficial layers of PFC[48,62], and the properties of vH input across these layers is distinct, with different contributions of direct excitation and feedforward inhibition[62]. Consistent with this it is interesting to note that there seems to be specific targeting by vH input to PFC neurons projecting to unique downstream areas[46]. How the role of vH input is differentiated across layers and projection populations in PFC is an interesting future direction.

**The contribution of vH-PFC neurons to approach and avoidance decisions**. We found that the two layers of vH-PFC neurons exhibited opposing changes in their activity upon open and closed arm entry in the EPM (Fig. 5). However, it is interesting to note that this activity was predominantly after entry to the arm, and hence potentially after the decision to enter the arm has been made. This is consistent with the majority of activity seen in PFC and vH during EPM exploration where specificity in neuronal activity is predominantly observed after arm entry[4,12,18,60]. However, this does not immediately fit with the previously described key role for this circuit in the decision to enter the open or closed arms[4,9,11,19–21,48], and also our optogenetic data showing that activation of each layer could bidirectionally bias entry to either the open or closed arms (Fig. 6).

One explanation for this discrepancy might be that vH-PFC neurons simply define whether the arm is open or closed once the mouse has entered the arm (or more generally whether the mouse is currently approaching or avoiding potential adversity[4,12,18,60]). This fits well with the idea of the hippocampus providing information about 'state', with subsequent decisions based on this information carried out in PFC and other downstream regions[64]. In this scenario, artificial activation of superficial vH-PFC activity may drive exploration or approach, while deep layer activation may drive a more general state of avoidance or exploitation— consistent with its collateral projections to NAc and LH[4,49,65].

Importantly, we did observe evidence for increases in activity of deep layer vH-PFC neurons immediately before arm entry (Fig. 5g, h). This might suggest that deep layer neurons are uniquely involved in the decision process, while superficial layer neurons are preferentially involved in representation of the maze – consistent with the preferential connectivity of this layer with entorhinal cortex. However, this preemptive activity of deep vH-PFC neurons did not differ across the open and closed arms. This may be due to the photometry approach employed to enable simultaneous recording from both populations. Both vH and PFC are structured to represent external and internal variables using a complex, single cell code[66], and it is difficult to accurately relate bulk activity measured using photometry to the activity of individual neurons. Therefore, an

important future direction will be to investigate the contribution of individual vH-PFC neurons in each layer.

Finally, this discrepancy could also be due to the inherent disadvantage of the way the classic EPM task is carried out - where decisions and internal variables are inferred by the location of the mouse within the maze. Due to this issue, it is difficult to precisely identify the timing of decision variables. For example, in the EPM it is presumed that the decision to enter the open arm is made just before entry[7,48,60], when in fact this choice could be made over a large period of time ranging from many seconds before arm entry to when the mouse is already in the proximal parts of the chosen arm. On this note it is important to mention that activity of both layers of vH-PFC neurons was not limited to arm entries (Fig. 5d), and the activity of vH-PFC neurons was not constant over consecutive arm entries (Supplementary Fig. 10), suggesting many more variables are represented by these neurons than we describe here. Therefore, an important future direction will be to investigate the activity of the two vH-PFC layers in behavioural tasks where the learning and updating of different stages of the decision process can be accurately separated[67–69].

**Necessity and sufficiency of vH-PFC neurons during EPM exploration**. We found that optogenetic activation of vH terminals in PFC from each layer could bidirectionally alter behaviour on the EPM (Fig. 6). Superficial layer activation promoted exploration of the open arms, while deep layer activation inhibited exploration. However, it is important to note that there are a number of technical considerations that must be taken into account when interpreting these experiments. First, terminal stimulation using optogenetics is not physiological, and can cause antidromic spiking. This could result in activity in areas with collateral projections from vH-PFC neurons such as NAc and LH. Although neurons with such collaterals were relatively rare in vH (~5%, Supplementary Fig. 2), we found that these neurons were found preferentially in the deep layers. Interestingly, antidromic spiking in this projection is thought to be very rare[11], however, it is unknown what effect such collateral projections might play. Second, while our optogenetic experiments show that each pathway is sufficient to change behaviour in the EPM, they do not show that the activity of these pathways is necessary to support such behaviour. There are a number of caveats that make such experiments difficult to interpret. For example, terminal manipulation using the majority of inhibitory opsins has large, off-target effects that often result in opposite physiological effects than those intended[70–72], while inhibition using retrograde infection is often inefficient, and confounded by collateral projections to distinct downstream regions[12,31] (see Supplementary Fig. 2). However, recently developed G-protein coupled optogenetic constructs - which lack the caveats of classic optogenetic tools when used at terminals - may provide a means to carry out these experiments for the first time[73,74]. Therefore an important future direction is to investigate the necessity of the two pathways during approach avoidance conflict.

Overall, our findings provide a synaptic and circuit mechanism for the regulation of behaviour during approach avoidance conflict: through two specialized parallel circuits located in the CA1/subiculum border of ventral hippocampus that allow bidirectional control of PFC. With this in mind, disruption in the balance of excitation and inhibition in PFC is closely associated with the transition to mental illness[75]. Alterations in vH input to PFC are thought to be key for this disruption, in particular in response to chronic stress and genetic mutations associated with schizophrenia[20,76,77]. Our study therefore reveals a mechanism by which changes in the activity of each layer in vH could exert strong shifts in excitatory and inhibitory balance in

PFC. Thus, understanding how these two layers may be differentially impacted in models of mental illness is an interesting future avenue of investigation.

## Methods

**Animals**. In total, 6–10 week old (adult) male and female C57 / bl6J mice provided by Charles River were used except where noted. To target inhibitory neurons we used the *Slc32a1(VGAT)-IRES-Cre* (#016962) knock-in line. To target Calbindin expressing neurons we used the *Calb1-IRES-Cre* (#028532) knock-in line. To target parvalbumin positive interneurons we used the *PV-IRES-Cre* (#008069) knock-in line. All were obtained from Jackson laboratories and bred in-house. Mice were housed in cages of 2–4 and kept in a humidity- (45–60%) and temperature- (20–24 °C) controlled environment under a 12 h light/dark cycle (lights on 7 am to 7 pm) with ad-libitum access to food and water. All experiments were approved by the U.K. Home Office as defined by the Animals (Scientific Procedures) Act, and by the University College London Animal Welfare Ethical Review Body.

### Stereotaxic surgery
*Retrograde tracers*. Red and green fluorescent retrobeads (Lumafluor, Inc.) for electrophysiological recordings.

Cholera toxin subunit B (CTXβ) tagged with Alexa 555 or 647 (Molecular Probes) for histology experiments.

*Viruses.*

| | |
|---|---|
| AAV2/1-CaMKII-GFP | (a gift from Edward Boyden; Addgene plasmid #64545) |
| AAVretro-Ef1a-DO_DIO-TdTomato_EGFP | (a gift from Bernardo Sabatini; Addgene plasmid #37120, see[34]) |
| AAV2retro-CAG-Cre | (UNC vector core) |
| AAV2/1-CaMKii-Cre | (UNC vector core) |
| AAV2/1-synP-FLEX-split-TVA-EGFP-B19G | (a gift from Ian Wickersham; Addgene plasmid #52473, see[78]) |
| EnvA-ΔG-RABV-H2B-mCherry-2A-CLIP | (a gift from Marco Tripodi, LMB, Cambridge, UK) |
| AAV2/1-EF1a-FLEX-hChR2(H134R)-EYFP | (a gift from Karl Deisseroth; Addgene viral prep #20298-AAV1) |
| AAV2/1-hSyn-hChR2(H134R)-EYFP | (a gift from Karl Deisseroth; Addgene viral prep #26973-AAV1) |
| AAV2/1- EF1a-FAS-hChR2(H134R)-EYFP | (a gift from Bernardo Sabatini; based on Addgene plasmid #37090, see[34]) |
| AAV2/9-mDlx-NLS-mRuby2 | (a gift from Viviana Gradinaru; Addgene viral prep #99130-AAV1, see[79]) |
| pAAV2/9-hSynapsin-FLEX-soCoChR-GFP | (a gift from Edward Boyden, Addgene viral prep #107712-AAV9, see[42]) |
| AAVretro.Syn.Flex.NES-jRGECO1a.WPRE | (a gift from Douglas Kim & GENIE Project; Addgene viral prep #100853- AAVrg, see[80]) |

*Surgery*. Stereotaxic injections were performed on 7–10 week old mice anaesthetized with isoflurane (4% induction, 1–2% maintenance) and injections carried out as previously described[40,81]. Briefly, the skull was exposed with a single incision, and small holes drilled in the skull directly above the injection site. Injections were carried out using long-shaft borosilicate glass pipettes with a tip diameter of ~10–50 μm. Pipettes were back-filled with mineral oil and front-filled with ~0.8 μl of the substance to be injected. A total volume of 140–160 nl of each virus was injected at each location in ~14 or 28 nl increments every 30 s. If two or more substances were injected in the same region they were mixed prior to injection. The pipette was left in place for an additional 10 min to minimize diffusion and then slowly removed. If optic fibres were also implanted, these were inserted immediately after virus injection ~300 μm above the area intended for recording or manipulation, secured with 1–2 skull screws and cemented in place with C&B superbond. Injection coordinates were as follows (mm relative to bregma):

| | | | |
|---|---|---|---|
| infralimbic PFC: | ML: ± 0.4, | RC: + 2.3, | and DV: - 2.4, |
| Diagonal Band of Broca: | ML: ± 0.2, | RC: + 0.7, | and DV: - 4.0, |
| Anterior Thalamus: | ML: ± 0.2, | RC: - 0.4, | and DV: - 3.2, |
| CA3: | ML: ± 2.8, | RC: - 3.0, | and DV: - 4, |
| Entorhinal Cortex: | ML: ± 2.7, | RC: - 5.2 (@ 10°), | and DV: - 4.5, |
| ventral Hippocampus: | ML: ± 3.4, | RC: - 3.7, | and DV: - 4.3. |

After injection, the wound was sutured and sealed, and mice recovered for ~30 min on a heat pad before they were returned to their home cage. Animals received carprofen in their drinking water (0.05 mg / ml) for 48 hrs post-surgery as well as subcutaneously following surgery (0.5 mg / kg). Expression occurred in the injected brain region for ~2 weeks until preparation of acute slices for physiology experiments, or fixation for histology. The locations of injection sites were verified for each experiment.

### Anatomy
*Pseudotyped rabies labelling from PFC*. For pseudotyped rabies experiments two injections were necessary. First, *AAV2/1-synP-FLEX-split-TVA-EGFP-B19G* was injected into the PFC of either *VGAT-Cre* mice to express TVA and G protein in inhibitory neurons, or was coinjected in a 1:1 mixture with *AAV2/1-CaMKii-Cre* into Cre negative littermates to label excitatory neurons. In addition, CTXβ was coinjected into PFC to achieve non-specific retrograde labelling for comparison across mice. 2 weeks later, 200 nl of *EnvA-ΔG-RABV-H2B-mCherry* was injected into PFC to infect and induce transynaptic spread only from neurons expressing Cre. Animals were prepared for histology after 7 days of rabies-mediated expression.

*TRIO labelling from PFC-projecting vH neurons*. TRIO labelling[82] again required two surgeries. First, *AAVretro-CAG-Cre* was injected into the PFC to express Cre recombinase in neurons projecting to PFC. In the same surgery *AAV2/1-synP-FLEX-split-TVA-EGFP-B19G* was injected into the vH to express rabies TVA and G-protein only in vH neurons that express Cre (i.e. only those that project to PFC). 2 weeks later, 200 nl of *EnvA-ΔG-RABV-H2B-mCherry* was injected into vH to infect and induce trans-synaptic spread only from PFC-projecting vH neurons. Animals were prepared for histology after 7 days of rabies-mediated expression.

*Histology*. Mice were perfused with 4% PFA (wt / vol) in PBS, pH 7.4, and the brains dissected and postfixed overnight at 4 °C as previously described[40]. In summary, 70 μm thick slices were cut using a vibratome (Campden Instruments) in either the transverse, coronal or sagittal planes as described in the figure legends. Slices were mounted on Superfrost glass slides with ProLong Gold or ProLong Glass (for visualization of GFP) antifade mounting medium (Molecular Probes). NucBlue was included to label gross anatomy. Imaging was carried out with a Zeiss Axio Scan Z1, using a 10x air immersion lens and standard filter sets for excitation/emission at 365-445/50 nm, 470/40-525/50 nm, 545/25-605/70 nm and 640/30-690/50 nm. Raw images were exported using Zen software (Zeiss) and analyzed with FIJI.

*Immunostaining*. Brain slices (70 μm thick) were prepared as above and incubated in blocking solution (3% bovine serum albumin, 0.5% triton in PBS) for 3 h at room temperature with constant agitation. After blocking, slices were incubated overnight at 4 °C in blocking solution containing either 1:5000 anti-DDDDK (ab1257, Abcam) to label smFP, or 1:1000 anti-calbindin (D-28k, Swant). Slices were then washed 3 times for 30 min each wash in PBS before incubation with 1:500 secondary antibody in blocking solution for 4 hours at room temperature (either Alexa 555- or Alexa 647-conjugated donkey anti-goat, A21432 / A21447, Invitrogen—to label smFP; or either Alexa 555- or Alexa 488-conjugated donkey anti-mouse, A31570 / A21202, Invitrogen—to label calbindin). Slides were mounted after a further 3 washes in PBS as above.

*Analysis of spatial distribution of PFC-projecting hippocampal neurons*. The spatial distribution of retrogradely labelled PFC-projecting hippocampal neurons was analyzed in transverse slices from mice injected with cholera toxin in PFC, and imaged as described above. In total, 3 70 μm slices spanning 350 μm between ~−3.5 and −5.0 mm DV were analyzed per injection. Sections containing the hippocampus were straightened along the cell body layer using the *straighten* macro in FIJI to produce a single field stretching from proximal CA3 to distal subiculum. Labelled cells within each slice were manually counted, and the coordinates saved for later analysis. The majority of cells were found at the CA1 / subiculum border - defined as the point where the hippocampal cell layer widens into a more subicular-like structure, which occurs consistently at ~0.7 of the distance of the entire straightened field. Custom scripts written in Python based around the sci-kit.learn package were used to cluster the coordinates using a Gaussian Mixture Model (GMM). Models containing between 1 and 6 components were fit to the data, and Bayesian Information Criterion (BIC) was used to select the best fit.

*Analysis of rabies labelling from VGAT and CaMKii neurons in PFC*. As above, transverse hippocampal slices using cholera toxin and rabies labelling of PFC-projecting cells were used for cell counting. All cells in the straightened hippocampal formation were counted, irrespective of fluorescent label to gain an overall distribution of vH-PFC neurons. These coordinates were clustered using GMM as above, where again, all analyzed fields were best fit by two components. Rabies positive cells were assigned to one of the two components (i.e., superficial and deep layers) created by the clustering algorithm. Data is presented as the proportion of all rabies labelled cells in each layer. As an internal control data for cholera toxin labelling is also presented using the same method, which shows there are no differences in the overall layer structure between the *CaMKii* and *VGAT* experiments.

*Whole-hippocampus cell quantification and analysis*. For quantification of CTXβ-labelled hippocampal neurons projecting to PFC, consecutive 60 μm coronal

sections spanning the hippocampus (approximately AP −4.3 mm to −1.0 mm) were collected from 4 independent injections. Cell counting of retrogradely labelled neurons was conducted using custom written scripts in R based around the WholeBrain package[30], an automatic segmentation and registration workflow in R. Segmentation was performed using wavelet multiresolution decomposition on the WholeBrain platform in R. To achieve accurate segmentation, a binary black and white mask of the position of every cell in the hippocampal slice was used. For the registration of coronal sections, six to eight random brain sections were sampled and manually annotated with approximate AP coordinates (by comparing to the Allen Brain Reference Atlas), and the remaining sections were assigned AP coordinates based on interpolation. Subsequently, registration was conducted on all images in a semi-automated manner; this involved an automatic registration of the tissue section by the WholeBrain software, and then refining the registration through manually adding, subtracting or changing of correspondence points at clear anatomical landmarks. Cell count data were saved as RData files, imported into Python and analyzed using Python 3.6. Only neurons residing in the ventral portion of the brain (DV < −3.5) were analyzed.

**Electrophysiology**

*Slice preparation.* Hippocampal recordings were studied in acute transverse slices, while prefrontal cortical recordings in Fig. 4 were studied in acute coronal slices. Mice were anaesthetized with a lethal dose of ketamine and xylazine, and perfused intracardially with ice-cold external solution containing (in mM): 190 sucrose, 25 glucose, 10 NaCl, 25 NaHCO$_3$, 1.2 NaH$_2$PO$_4$, 2.5 KCl, 1 Na$^+$ ascorbate, 2 Na$^+$ pyruvate, 7 MgCl$_2$ and 0.5 CaCl$_2$, bubbled with 95% O$_2$ and 5% CO$_2$. Slices (300 μm thick) were cut in this solution and then transferred to artificial cerebrospinal fluid (aCSF) containing (in mM): 125 NaCl, 22.5 glucose, 25 NaHCO$_3$, 1.25 NaH$_2$PO$_4$, 2.5 KCl, 1 Na$^+$ ascorbate, 3 Na$^+$ pyruvate, 1 MgCl$_2$ and 2 CaCl$_2$, bubbled with 95% O$_2$ and 5% CO$_2$. After 30 min at 35 °C, slices were kept for 30 min at 24 °C. All experiments were conducted at room temperature (22–24 °C). All chemicals were from Sigma, Hello Bio or Tocris.

*Whole-cell electrophysiology.* Whole-cell recordings were made from hippocampal pyramidal neurons retrogradely labelled with retrobeads or CTXβ, which were identified by their fluorescent cell bodies and targeted with Dodt contrast microscopy, as previously described[40,81]. For sequential paired recordings, neurons were identified within a single field of view at the same depth into the slice. The recording order was counterbalanced to avoid any potential complications that could be associated with rundown. For current clamp recordings (Fig. 2, Fig. 3a–d, Supplementary Fig. 5e, f), borosilicate recording pipettes (2–3 MΩ) were filled with (in mM): 135 K-gluconate, 10 HEPES, 7 KCl, 10 Na-phosphocreatine, 10 EGTA, 4 MgATP, 0.4 NaGTP. For voltage clamp experiments, three internals were used, First, in Figs. 3i and 4h, i, a Cs-gluconate based internal was used containing (in mM): 135 Gluconic acid, 10 HEPES, 7 KCl, 10 Na-phosphocreatine, 4 MgATP, 0.4 NaGTP, 10 TEA and 2 QX-314. In Fig. 3i excitatory currents were isolated by recording at − 70 mV in 10 μM extracellular Gabazine. In Fig. 4h, i, excitatory and inhibitory currents were electrically isolated by setting the holding potential at −70 mV (excitation) and 0 mV (inhibition) and recording in the presence of 10 μM extracellular APV. Second, to allow characterization of interneuron intrinsic properties, experiments in Fig. 4j–m were carried out using current clamp internal and excitatory currents were isolated using 10 μM extracellular Gabazine. Finally, to record inhibitory currents at rest we used a high chloride internal (in mM): 135 CsCl, 10 HEPES, 7 KCl, 10 Na-phosphocreatine, 10 EGTA, 4 MgATP, 0.3 NaGTP, 10 TEA and 2 QX-314, with 10 μM external NBQX and 10 μM external APV. For two-photon experiments, the internal solution also contained 30 μM Alexa Fluor 594 (Molecular probes). Recordings were made using a Multiclamp 700B amplifier, with electrical signals filtered at 4 kHz and sampled at 10 kHz.

*Viral strategy for in vitro optogenetics.* Presynaptic glutamate release was triggered by illuminating ChR2 in the presynaptic terminals of long-range inputs into the slice, as previously described[40,81]. Wide-field illumination was achieved via a 40 x objective with brief (0.2, 0.5, 1, 2 and 5 ms) pulses of blue light from an LED centred at 473 nm (CoolLED pE-4000, with appropriate excitation-emission filters). Light intensity was measured as 4–7 mW at the back aperture of the objective and was constant between all cell pairs.

To achieve afferent specific terminal stimulation in vH (Fig. 3a–d), AAV2/1-hSyn-hChR2(H134R)-EYFP was injected into each downstream area, and retrobeads were injected into PFC to label the two layers in vH.

To achieve input-specific terminal labelling and stimulation in PFC (Fig. 4), we injected AAV2/1-EF1a-FLEX-hChR2(H134R)-EYFP (creON) or AAV2/1-EF1a-FAS-hChR2(H134R)-EYFP (creOFF) into *Calb1-IRES-Cre* mice to target superficial neurons, and deep layer neurons respectively.

To target whole cell recordings to interneurons (Figs. 3i, 4j–m), we used AAV2/9-dlx-mRuby injections in the area of interest.

To express ChR2 in local PV + interneurons within vH (Fig. 3e, f), we injected AAV2/1-EF1a-FLEX-hChR2(H134R)-EYFP into the vH of *PV-IRES-Cre* mice, and retrobeads into PFC to allow paired recordings from the two layers.

To allow single cell optogenetic stimulation (Fig. 3g–i), we used an injection of AAVretro-syn-Cre in PFC, and an injection of AAV2/1-EF1a-DIO-SoCoChR2 in vH. The resultant sparse labelling of neurons across each layer allowed stimulation

of individual SoCoChR expressing neurons with a focused light spot (with an activation resolution of ~50 μm Fig. 3h, Supplementary Fig. 7), while recording from neighboring interneurons labelled with dlx-mRuby as above.

*2 photon imaging and image reconstruction.* Two-photon imaging was performed with a customized Slicescope (Scientifica), based on a design previously described[40,81]. For two-photon imaging, 780 nm light from an Erbium fibre laser (Menlo Systems) was used to excite Alexa Fluor 594. For each experiment a high-resolution stack of the entire neuron was taken for reconstruction of dendrite morphology.

*Electrophysiology data acquisition and analysis.* Two-photon imaging and physiology data were acquired using National Instruments boards and SciScan (Scientifica) and WinWCP (University of Strathclyde) respectively. Electrical stimulation was via a tungsten bipolar electrode (WPI) and an isolated constant current stimulator (Digitimer). Optical stimulation was via wide field irradiance with 473 nm LED light (CoolLED) as described above. Data was analyzed using custom routines written in Python 3.6. Current step data was analyzed using routines based around the Neo and eFEL packages. For synaptic connectivity experiments, amplitudes of PSPs were taken as averages over a 1-ms time window around the peak. For connectivity analysis, a cell was considered connected if the light-induced response was greater than 6 times the standard deviation of baseline.

Two-photon image reconstruction and analysis was carried out using VIAS and NeuronStudio[83], before further analysis was carried out using custom scripts in Python 3.6 based on the NeuroML package from the Human Brain Project. The online Human Brain Project morphology viewer was used for visualizing reconstructed neurons.

**Behaviour.** After sufficient time for surgical recovery and viral expression (>4 weeks), mice underwent multiple rounds of habituation. Mice were habituated to the behavioural testing area in their home cage for 30 min prior to testing each day. Mice were habituated to handling for at least 3 days, followed by 1–2 days of habituation to the optical tether in their home cage for 10 min.

*Bulk calcium imaging using fibre photometry during elevated plus maze exploration. Labelling superficial and deep vH neurons for photometry.* To allow simultaneous recordings of both superficial and deep vH-PFC populations, AAVretro-DIO-RGeCO1a (creON) and AAVretro-FAS-GCaMP6f (creOFF) were co-injected into PFC of *Calb1-IRES-Cre* mice as a 50:50 mix, and a 2.5 mm ferrule containing a 200 μm optical fibre was implanted in vH. This strategy allowed for simultaneous dual color imaging, as the red sensor RGeCO1a is expressed in cre-positive (*Calb1* +) superficial neurons, while the green sensor GCaMP6f is expressed in cre-negative (*Calb1*-) deep neurons in vH. Thus, separation of signals from the two layers is based on *Calb1* expression. FAS was used to counteract known interference between different Cre-dependent viruses[34].

*Bulk calcium imaging during behaviour.* After habituation (above), mice were exposed to the elevated plus maze (EPM) for 9 min, and allowed to freely explore the open and closed arms of the maze. To record calcium activity, we used a system based on[84–86]. Briefly, two excitation LEDs (565 nm 'green' and 470 nm 'blue') were controlled via a custom script written in LabView (National Instruments). Blue excitation was sinusoidally modulated at 210 Hz and passed through a 470 nm excitation filter while green excitation was modulated at 500 Hz and passed through a 565 nm bandpass filter. Excitation light from each LED was collimated, then combined using a 520 nm long-pass dichroic mirror. The excitation light was coupled into a high-NA (0.53), low-autofluorescence 200 μm patch cord by reflection with a multiband dichroic mirror and fibre coupler. Each LED was set to 100 μW at the far end of the patch cord, which was terminated with a 2.5 mm ferrule. The emission signal was collected through the same patch cord and collimator, and separated from the excitation light by the multiband dichroic. Green and red signals were split using a longpass dichroic mirror before being passed through a GFP emission filter and RFP emission filter respectively. Filtered emission was then collected by a femtowatt photoreceiver (Newport). The photoreceiver signal was sampled at 100 kHz, and each of the two modulated signals generated by the two LEDs was independently recovered using standard synchronous demodulation techniques implemented in real time by a custom Labview script (Supplementary Fig. 9). Three separated signals; green, red and autofluorescence (the signal from each sensor at the modulation frequency that does not match the sensor), were then downsampled to 50 Hz before being exported for further analysis. To account for gradual bleaching, a 3rd order polynominal fit was subtracted from each trace. After this, a least-squares linear fit was applied to the autofluorescence signal to align it to the green and red signals. Time series data was then calculated for each behavioural session as the zscore of (green or red signal—fitted autofluorescence signal, Supplementary Fig. 9). Calcium signals were sorted as 12 s epochs around behavioural events (rear, stretch attend, head dip, open arm entry, closed arm entry) scored by an expert observer (such that each event there was 6 s before the beginning of the behaviour, and 6 s after). For each event we subtracted baseline activity at the start of the trace ('bl', -6 to −4 s). We then analyzed activity immediately before the behavioural event ('pre', −2 to 0 s) and a timepoint after the initiation of the behavioural event that coincided with peak changes in activity of one or both layers (2 to 4 s for open arm

entry, 0 to 4 s for closed arm entry, 2 to 4 s for all other events). To investigate long term changes in activity (Supplementary Fig. 11a) we sorted traces for each layer into longer, 20 s epochs around open and closed arm entry, and also the point at which the mouse reached the farthest most point of the arm after each arm entry. We baselined both traces to the same point ($-10$ to $-8$ s before arm entry), and thus these traces show how activity evolves over the period of the entire arm. To investigate how activity around open arm entry changed across successive entries (Supplementary Fig. 11b, c), we classified arm entries for each mouse as either the first, middle (all entries bar the first and last), and last in the session. One mouse was excluded from this analysis as it only entered the open arms once. All optical components were from Thorlabs, Semrock or Chroma, patch cords were from Doric Lenses.

*Optogenetic manipulation of neurons during elevated plus maze exploration.* Axon terminals in PFC from superficial and deep vH neurons were labelled using a creON and creOFF approach as described above (see Optogenetics), and a 200 µm optical fibre was implanted unilaterally above right or left PFC in a counterbalanced manner. Mice expressing GFP in vH were used as controls. After habituation (above), behaviour was assessed using the EPM. As above, EPM sessions lasted 9 min. For stimulation, the session was split into 3, 3 min epochs (Fig. 6c). In total, 20 Hz light stimulation was delivered via a 473 nm laser, coupled to a patch cord (6-8 mW at the end of the patch cord) during the second three-minute epoch (3–6 min period of the test) to activate ChR2 positive vH terminals in PFC. Real-time light delivery was based on the location of the mouse in the EPM apparatus, where light stimulation occurred only during points where the mouse was in the centre or open arms of the maze[4,48]. Overall, light was only delivered during the middle 3 min epoch of the test, and only when the mouse entered the central zone of the maze. Stimulation continued as long as the mouse remained in the centre or open arms. Time spent investigating the open arms of the maze was scored for each mouse in each epoch using automated analysis (Deeplabcut[87]). We also performed a control experiment on a separate cohort of mice (Supplementary Fig. 13) where we continuously delivered 20 Hz light during the entire 3 -6 min stimulation epoch, irrespective of the position of the mouse. Open arm exploration was defined as entry of the head into the open arm, so as to include stretch attend, prospective exploration. To avoid ceiling effects in optogenetic behavioural experiments, mice were excluded if they failed to enter either open arm in the first three minutes of testing. This exclusion criterion was preestablished before the start of the experiment (see[88]). We also included an exclusion criterion where if mice did not enter the centre zone during the stimulation epoch (i.e. were never stimulated) they would be excluded. However, all mice passed this criterion and were included in the final analysis.

**Statistics**. Summary data are reported throughout the figures either as boxplots, which show the median, 75th and 95th percentile as bar, box and whiskers respectively, or as line plots showing mean ± s.e.m. Example physiology and imaging traces are represented as the mean ± s.e.m across experiments. Data were assessed using statistical tests described in the figure legends and supplementary statistics summary using the pingouin package in python. Significance was defined as $P < 0.05$, all tests were two sided. Multiple comparisons were corrected with a Holm-Šidák correction. No statistical test was run to determine sample size a priori. The sample sizes we chose are similar to those used in previous publications. Animals were randomly assigned to a virus cohort (e.g., ChR2 versus GFP), and where possible the experimenter was blinded to each mouse's virus assignment when the experiment was performed. This was sometimes not possible due to e.g. the presence of the injection site in the recorded slice.

**Reporting summary**. Further information on research design is available in the Nature Research Reporting Summary linked to this article.

## Data availability
Source data are provided for Figs. 1–6, and are available with this paper, and at https://github.com/macaskill-lab. Due to volume, raw data are available from the corresponding author (a.macaskill@ucl.ac.uk) upon reasonable request. Data from the Allen reference atlas (https://mouse.brain-map.org/static/atlas) was used in this study. Source data are provided with this paper.

## Code availability
Custom photometry acquisition code is available at https://github.com/macaskill-lab.

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

## Acknowledgements

We thank Marco Tripodi for providing rabies virus, Francesca Cacucci for help with surgery, and Gemma Estrada Girona and Marcus Stephenson-Jones for virus production. We thank Neil Burgess, Tara Keck, Thomas Mrsic-Flogel, Aman Saleem, Marcus Stephenson-Jones, Tom Wills, and members of the MacAskill laboratory for helpful comments on the manuscript. We also thank David Attwell, Francesca Cacucci, Mark Farrant, Troy Margrie and Andreas Schaefer for comments on a previous version. A.F.M. was supported by a Sir Henry Dale Fellowship jointly funded by the Wellcome Trust and the Royal Society (grant number 109360/Z/15/Z) and by a UCL Excellence Fellowship. C.S.B. was supported by the Wellcome Trust 4-year PhD in Neuroscience at UCL (grant number 206074/Z/17/Z). R.A. was supported by a King Fahad Medical City Studentship. K.M. was supported by the Wellcome Trust 4-year PhD in Neuroscience at UCL (grant number 215165/Z/18/Z). R.W.S.W. was supported by a UCL Graduate Research Scholarship and a UCL Overseas Research Scholarship.

## Author contributions

Conceptualization, C.S.B. and A.F.M.; Methodology, C.S.B. and A.F.M.; Investigation, C.S.B., R.A., K. M., R.W.S.W. and A.F.M.; Formal Analysis, C.S.B. and A.F.M.; Writing –

Original Draft, C.S.B. and A.F.M.; Writing – Review & Editing, C.S.B. and A.F.M.; Funding Acquisition, C.S.B. and A.F.M.; Supervision, A.F.M.

## Competing interests

The authors declare no competing interests.
