## [Peer Review File · Nature Communications]

Two opposing hippocampus to prefrontal cortex pathways for the control of approach and avoidance behaviorREVIEWER COMMENTS

Reviewer #1 (Remarks to the Author):

In this ms, Sanchez-Bellot & MacAskill, explore the differential properties and behavioral correlates of ventral hippocampal projections to the prefrontal cortex (PFC) in mice. Using a palette of viral strategies, they examine the intrinsic and morphological properties of PFC-projecting neurons at the CA1/subiculum border, dissect their local and brain-wide connectivity and evaluate/modulate their ongoing activity while freely moving mice explore the elevated plus maze (EPM). According to their data, PFC-projecting superficial cells which are more strongly innervated by the entorhinal cortex preferentially target PFC interneurons and favor open arm exploration. In contrast, PFC-projecting deep cells which are more influenced by septal and anterior thalamic inputs, target both PFC interneurons and pyramidal cells and favor open arm avoidance. These data provide a compelling assessment of the role and mechanisms of the ventral hippocampus to PFC pathway.

This is a very interesting paper with exciting new results. However, the current version lacks supporting material to validate my enthusiasm. Thus, most of my comments are focused on clarifications and demands for additional information to substantiate/support some of the strategies/analyses/results.

Major

1- The exact nature of PFC-projecting cells is unclear. The region from where these cells are examined corresponds to the distal aspect of the ventral CA1 at the border with the subiculum. According to the Hippocampus Gene Expression Atlas (Bienkowski et al., 2018) and cell-type specific gene mapping (elsewhere) these cells are not necessarily CA1 but a mix of distal CA1 and subicular cells. The location and distribution shown in Fig.1b and their bursting/non-bursting phenotype (Fig.1e,f) are more consistent with subicular cells or at least of cells recorded at the border. Actually, the segregated distribution of deep and superficial cells in Fig.1b,i reminds those subicular layers 2 and 4 (according to HGEA), and not the typical deep/superficial continuum of packed CA1. Making horizontal slices complicates interpretation due to the different spatial orientation of the different hippocampal axes and alignment with validated regions is not so straightforward. Thus, it would be very helpful to validate the CA1 or subicular nature of PFC-projecting cells by using immunostaining (IH) and/or any other complementary assay. While I don't feel this as a limiting factor, the whole paper would benefit from being more rigorous in the real nature of these cells.

2- Authors use Calb-cre mice to target superficial CA1 cells. While this is the case for the dorsal hippocampus, it is unclear how calb expression relates to superficial CA1 cells at more ventral regions. It is also unclear how calb is expressed in the subiculum. In addition, authors should provide validation of their Calb-Cre on/off strategy. Finally, calbindin is also expressed in interneurons and in dentate granule cells and authors need to exclude any potential contribution/contamination. For interneurons in particular, calbindin expressing interneurons may be projecting cells. This issue is relevant as many of the viral strategies uses ubiquitous promoters. Thus, overall, the paper lacks detailed anatomical validation and support.

3- Authors use fiber photometry to evaluate activity of PFC-projecting deep and superficial cells during EPM exploration. While most analysis concentrate in evaluating transient events, the basal level of optical signal is not discussed nor shown. Only 4 sec of entry on the arm are analyzed so, it is unclear what is the typical activity along the entry and how does it evolve over exploration. Raw data and validation of a stable basal level should be provided as supplementary information. It is also unclear whether the fiber is sensing bulk activity from dispersed soma (as suggested in previous figures) and how precisely can deep and superficial cells be separated given their different density (and hence contribution to bulk imaging). Are fibers of passage or dendrites of deeper cells contaminating superficial signals? In addition, how can authors precisely define the entry of mice into the arm? All these experiments require substantial support from the anatomical and analytic point of views.

4- Authors use optogenetics to manipulate the activity of deep and superficial PFC-projecting cells

while mice explores the EPM. I am not fully convinced the behavioral effects are clean and statistically strong. First, it is unclear how the protocol goes. Authors light on only during the 3 min period in the central area or in the open arms. What if an animal stays in the closed arm? What proportion of animals receive lights actually? Second, there is a lot of variability both in Cre-ON and GFP-control animals regarding the time in open arm, so it is unclear what the normalized time is doing here to reach significant statistical effects. All animals exhibit a decrease of exploration to the open arm (and hence they should be increasing exploration of the closed arm) along the task except for the CreON group, and some individual GFP-control mice. Thus, the effect of optogenetic looks a bit at the limit. Authors need to show convincingly that the behavioral effect is robust. Is stim in the closed arm exerting a complimentary effect that could reinforce statistics and conclusions?

Other comments

5- Regarding quantification in Fig.1: A) Please clarify whether quantification of radial distance in Fig.1b refers to several sections or is just one representative section; B) Please, clarify whether cells are counted in a stack (70 μ m thick) or at a plane; how many planes? What objective was used to quantify? C) In methods authors refer to "either the transverse, coronal or sagittal planes as described in the figure legends", but most examples look horizontal or sagittal. No coronal sections are presented which would facilitate clarification of my point 1 (HGEA are represented in coronal sections). D) Fig.1c and Supp.Fig.1d: authors refer to pia, but strictly speaking, this is the alveus. The pia is the layer of meninges surrounding the surface of the brain, which is more likely to be at the hippocampal fissure. E) Fig.1i, please improve contrast of the fluorescent image

6- Check arrowheads in sup.Fig.2b; some point erroneously to distinct cells.

7- Fig.2f: the retrobead injection is lacking. According to the text retrobead-negative cells are recorded here

8- Authors declare "Excitatory local connectivity in hippocampal CA1 and subiculum is rare" but this is not strictly correct. Not certainly at the ventral subiculum, nor between CA1 and subicular cells.

9- Experiments in Fig.2g,h,i exploits somatic ChR2 but no supporting material is provided to validate their resolution. In addition, can authors fully exclude the fibers of passage? Also validation of interneuron-specific expression of mRuby (and electrophysiological validation) should be provided. A supplementary figure is critical here

10- For the quantification of PFC-projecting cells in Fig.3a,b: it is unclear why authors present data as proportion of cells in the superficial layers and not along the normalized distance as they reported before. Interpreting this plot is very tricky. Also, in the table authors refer the n for deep and sup which unclearly relate to CamKII vs VGAT (as shown in Fig.3b).

11- Regarding axonal arborization of PFC-projecting cells in Fig.3c, authors need to show the distribution of the axonal plexus in the PFC to clarify whether deep and sup cells are actually innervating different layers and/or dendritic domains which could additionally complicate interpretation. Interestingly, authors record only from deep PFC cells, while Lee et al., Neuron 2019 (ref 44) record from layers II/III. Fig.3f: what PFC layers are recorded here? All this should be more carefully considered and justified.

12- Supp.Fig.3: the injection site and fiber placement: please, show some examples to appreciate the size and potential anatomical effect of the fiber implantation.

13- I strongly recommend separating fiber photometry and optogenetic behavioral experiments in different figures, and adding substantially more analysis, and support to results.

14- Depending on their ability to address my point 1 I would recommend authors making more clear the CA1/subicular nature of the PCF-projecting cells and shape title and abstract accordingly.

15- Discussion, in general reads a bit oversimplified. I would have several comments and additions but in view of the overall assessment and the many details authors need to work out to improve this interesting paper I would save my comments for a revised version.

Reviewer #2 (Remarks to the Author):

In their manuscript, Sánchez-Bellot and MacAskill examine the local- and long-range connectivity of two circuits connecting the ventral hippocampus to the prefrontal cortex, which differ based on radial depth of ventral hippocampus neurons, and relate these circuits to approach/avoidance behavior. They associate activity in the deep ventral hippocampus circuit with avoidance behavior and activity in the superficial ventral hippocampus circuit with exploration/approach. Finally, they show that optogenetic activation of each of these circuits leads to the predicted behavior, demonstrating a causal mechanism by which the hippocampus plays a bidirectional role in approach/avoidance behavior.

The experiments presented in the manuscript are challenging and well-executed. The paper does an outstanding job of bridging current coarse scientific knowledge into fine-scale discovery: the authors hinge their experiments on previously identified cell types (i.e., superficial vs. deep ventral hippocampus neurons) and a known bidirectional role for the hippocampus in approach/avoidance, but are the first to show that this hippocampus function derives from distinct circuits involving these cell types. As such, the paper fills a clear knowledge gap and is timely. Given their novel findings spanning multiple scales (i.e., circuit mapping to behavior), recapitulation and extension of previous results, and the potential eventual translational relevance of their results, this manuscript is of interest to broad audience and well-suited to a strong journal like Nature Communications.

In this reviewer's opinion, no major revisions nor further experiments are needed. However, the clarity and convincingness of some parts of the paper could be improved by considering the following minor comments and addressing via additional discussion or rewording.

Minor comments:

- CRACM-based conclusions: I would suggest that the authors exhibit more restraint in discussing their CRACM-associated conclusions for Fig. 2. Their recordings are not well-positioned to compare inputs across superficial vs. deep cells for at least two reasons. First, morphological differences will mean that inputs – especially distal inputs – will have differential somatic signatures. Second, recordings were performed in current clamp, which means that cell-intrinsic differences may distort synaptic inputs. Thus, more equivocal wording should be employed – claims like “additional input” can be suggested by not truly resolved by these experiments.
- Off-target effects, optogenetic stimulation: Stimulation of axon terminals can result in antidromic spiking back to the cell body, which can then orthodromically propagated to other axon terminals that are associated with different collateralized circuits. This off-target stimulation of collateralized circuitry can be a potential confound. The authors should discuss this possibility in the context of their experimental results.
- CA1 vs subiculum: While the authors state that the neurons they study are at the “CA1/subiculum” border, they do not attempt to describe which brain region (CA1 vs subiculum) these neurons belong to. Given that the exact boundary between the CA1 and subiculum has been debated in the field (especially in ventral hippocampus, as in this study), this is a challenging task. Nonetheless, some discussion as to what hippocampal region these cells reflect, or indeed if this is even possible to interpret the context of the authors' experiments, is important. As many in the community think about these outputs in terms of CA1 vs. subiculum, it will be important for this paper to discuss their results in the context of these regions.
- CreON vs. creOFF explanation: The wording describing the creON/creOFF experiment was confusing and seemed contradictory: “[We] injected an AAV into vH to express either ChR2 where expression was limited to cre-expressing (superficial layer) neurons (creON ChR2), or where ChR2 was inhibited in cre-expressing neurons, and thus was only expressed in cre-negative (deep layer) neurons,” as the sentence structure makes it seem as though in both creON and creOFF ChR2 is expressed in only cre-negative neurons.

- Figure legends: It would be helpful to have a figure legend that describes what the orange neuron in Figure 3c is (it is provided in 3f, but would make more sense to have this legend in the first panel the reader views). In the current text, it is also unclear how this schematized neuron is related to the experiment.
- Reporting of cell sample sizes: Often, the number of superficial vs. deep ventral hippocampus and PFC analyzed cells are not reported in the text or on relevant figures (e.g., Figure 1g—percentage of bursting cells is reported for both superficial and deep hippocampus cells, but not the actual number of cells examined; although there are dots for individual cells, the burden is on the reader to manually add the data points). Please report sample sizes throughout, as possible.

Reviewer #3 (Remarks to the Author):

Sanchez-Bellot & MacAskill used a combination of in vitro slice electrophysiology, anatomical tracing, in vivo fibre photometry and optogenetics to identify two populations of medial prefrontal cortex (mPFC)-projecting ventral hippocampal (vH) cells that have distinct electrophysiological and projection profiles, as well as opposing control over the expression of innate approach-avoidance conflict, as measured in the elevated plus maze (EPM). The major findings of the study are that vH cells in the superficial layers in the subiculum and CA1 receive medial entorhinal inputs, and connect to both somatic and dendritic inhibitory neurons in the PFC in order to facilitate approach/exploratory behavior. In contrast, vH cells residing in the deep layers receive subcortical projections (anterior thalamus, DBB) and project onto somatic interneurons and pyramidal cells in the PFC to promote avoidance behavior.

This is a very interesting set of experiments that have been conducted carefully and thoroughly. The findings provide novel mechanistic insights into the control of anxiety, and will likely be of interest to an audience interested in the neurobiology of anxiety, and function of hippocampal and prefrontal circuits. The statistical analyses are appropriate, and the methodology is clearly written and reproducible on the whole, although more clarification and details are warranted in some parts, as described below. I have also outlined some concerns that would need to be addressed in order for me to fully assess the validity of the claims.

I appreciate the reason why the authors chose to stimulate contralateral vCA3 inputs to the PFC projecting cells, but this raises the question of whether there may be intrinsic differences in the projection pattern/physiological profile of the neuronal population targeted by ipsilateral vs. contralateral vCA3 inputs, and whether the omission of recording from the ipsilateral pathway in the current study is an issue.

Did the authors assess the collateral projections of the superficial and deep layer PFC-projecting vH cells?

The behavioral and data analyses that the authors conducted are not best suited to justifying some of the claims the authors are making. For instance, the authors claim their fiber photometry findings “suggested that the relative activity of the superficial and deep layers of the vH-PFC projection around the choice point of the EPM may inform the decision to approach or avoid the open arms.” However, neural activity was analysed during 0 to 4 seconds after arm entry, and changes in activity also appear most prominent after the entry into either the open or closed arm (Figure 4 c,d). Given that approach and avoidance decisions likely occur before the animal enters the arm, and not after, it is possible that the time window the authors include in their analysis may not be representative of decision-making. Was neural activity analysed for the 2s preceding arm entry (-2s to 0s)?

Clarification is needed as to how exactly the $\Delta F/F$ values were obtained. The methods state “Time series data was then calculated for each behavioral session as the z-score of (green or red signal - fitted autofluorescence signal)” which accounts for the the z-score of ΔF shown in the traces. The authors also state, “Activity was analyzed as the area under the curve from 0 to 4 s after arm entry, normalized to the area under the curve -4 to -2 s before arm entry”. I assume this resulted in the reported $\Delta F/F$ values but it is not clear how these were derived.

Please provide details as to how many open and closed arm entries were included in the data analysis. Did the observed changes in activity differ between successive entries into the open or closed arms (those made at the beginning vs. end of session)?

It is of concern that the placements of optic fibres in the fibre photometry experiments seem to be much more posterior and dorsal to the sites targeted in the tracing and in vitro slice experiments (Supp Fig 3b).

The inclusion of the optogenetic activation experiment is a nice way of demonstrating the sufficiency of the differential vH-mPFC circuits in approach/avoidance behaviors. However, it is not a demonstration of the necessity of these circuits in the control of anxiety – did the authors examine the effects of inhibiting these circuits on approach/avoidance?

Perhaps the manuscript is intended to be a short report, but I found the discussion to be lacking, and have a few suggestions for improvements. Firstly, the authors should discuss their findings in the light of recent work that explore the contributions of different vH subfields in innate and cued approach-avoidance behaviors (e.g., PMID: 25931556, PMID: 29606418, PMID: 32929245). Secondly, it's not clear how exactly the reported circuit mechanisms (particularly the contrasting innervation patterns of the mPFC) give rise to the implementation of preferential approach vs. avoidance in a conflict situation. Furthermore, huge conceptual leaps are made in stating that superficial neurons use structured information from the cortex to plan and promote approach/exploratory behavior, and deep layer neurons use salient information in the environment flexibly to promote avoidance. The authors should better articulate/elaborate on how they have arrived at these conclusions.

Please indicate if, and which within subject post-hoc comparisons were significant in the EPM data (Figure 4 g, i)

Sanchez-Bellot and MacAskill: Response to reviewers' comments.

We thank all of the reviewers for their constructive comments. For ease, we have colour coded our response – the original comments are in black, our responses are in blue, and quoted text from the manuscript is in grey. In addition, we have highlighted changes to the manuscript in blue.

Reviewer #1 (Remarks to the Author):

In this ms, Sanchez-Bellot & MacAskill, explore the differential properties and behavioral correlates of ventral hippocampal projections to the prefrontal cortex (PFC) in mice. Using a palette of viral strategies, they examine the intrinsic and morphological properties of PFC-projecting neurons at the CA1/subiculum border, dissect their local and brain-wide connectivity and evaluate/modulate their ongoing activity while freely moving mice explore the elevated plus maze (EPM). According to their data, PFC-projecting superficial cells which are more strongly innervated by the entorhinal cortex preferentially target PFC interneurons and favor open arm exploration. In contrast, PFC-projecting deep cells which are more influenced by septal and anterior thalamic inputs, target both PFC interneurons and pyramidal cells and favor open arm avoidance. These data provide a compelling assessment of the role and mechanisms of the ventral hippocampus to PFC pathway.

This is a very interesting paper with exciting new results. However, the current version lacks supporting material to validate my enthusiasm. Thus, most of my comments are focused on clarifications and demands for additional information to substantiate/support some of the strategies/analyses/results.

We thank the reviewer for their positive assessment of our manuscript, and we hope that our additional experiments and analysis, outlined below, will mitigate their concerns.

Major

1- The exact nature of PFC-projecting cells is unclear. The region from where these cells are examined corresponds to the distal aspect of the ventral CA1 at the border with the subiculum. According to the Hippocampus Gene Expression Atlas (Bienkowski et al., 2018) and cell-type specific gene mapping (elsewhere) these cells are not necessarily CA1 but a mix of distal CA1 and subicular cells. The location and distribution shown in Fig.1b and their bursting/non-bursting phenotype (Fig.1e,f) are more consistent with subicular cells or at least of cells recorded at the border. Actually, the segregated distribution of deep and superficial cells in Fig.1b,i reminds those subicular layers 2 and 4 (according to HGEA), and not the typical deep/superficial continuum of packed CA1. Making horizontal slices complicates interpretation due to the different spatial orientation of the different hippocampal axes and alignment with validated regions is not so straightforward. Thus, it would be very helpful to validate the CA1 or subicular nature of PFC-projecting cells by using immunostaining (IH) and/or any other complementary assay. While I don't feel this as a limiting factor, the whole paper would benefit from being more rigorous in the real nature of these cells.

We thank the reviewer for their comments and agree that a more easily comparable dataset would be beneficial to allow better understanding of the different populations of vH-PFC neurons. This is especially true, as is noted by reviewer 2, as the boundaries of CA1 and subiculum in ventral hippocampus are not clear, and there is considerable debate as to the organisation of these ventral hippocampal subfields. This is most notable in comparison to the tight spatial structure of dorsal hippocampus where CA1, distal and proximal subiculum are discretely organised (e.g., in Cembrowski et al. 2019), while in ventral hippocampus this is not the case and neurons in the CA1 and subiculum areas seem to be organised as a gradient (Wee et al, 2020).

To address this issue, we have carried out a number of new experiments, and also added substantially to our results and discussion as outlined below:

We have repeated our initial retrograde tracing experiment, and have analysed coronal sections to allow such a comparison – presented in **Fig.1** and in **Sup.Fig.1**. We have registered these coronal sections across the entire hippocampal formation to the Allen Brain Atlas coordinate framework, which allows direct comparison to previous studies.

In agreement with our previous descriptions - and the reviewers' intuition - we found that vH-PFC neurons are present at the border between CA1 and subiculum – namely distal CA1 and proximal subiculum, consistent with many previous reports. We found that these neurons are spread in a continuous gradient across the proximal:distal axis, with equal proportions in both CA1 and subiculum.

This new registered dataset also allowed us to align labelled neurons to the HGEA atlas from Bienkowski et al 2018. Interestingly, at the distal extent of vH-PFC labelling, superficial layers aligned well with subiculum layer II, but deep layers did not align well with any layer, and instead were consistently present at the border between layers III and IV. We go on to note that the vH-PFC labelling is better described by an alternative description of subicular layers presented by Ishihara and Fukuda, 2016 (**Sup.Fig.1**).

Overall, we conclude that our data suggests that vH-PFC neurons in vH are organised as a continuum across the proximal distal axis encompassing both CA1 and subiculum. However, this distribution can be separated into two layers along that radial axis that correspond distally to superficial and deep layers of the subiculum.

We have presented this data in **Fig.1** and in **Sup.Fig.1**, and discussed it in a new section in the results starting on *p4*:

“Despite intensive investigation of the functional and behavioral properties of the vH to PFC projection, the cellular distribution of these neurons in hippocampus remains unclear. This is compounded by that fact that the precise boundary between the classical long-range output regions of hippocampus - distal CA1, proximal and distal subiculum is unclear and widely debated. To investigate the cellular organization of the projection from vH to PFC, we labelled neurons that project to the PFC with an injection of the retrograde tracer cholera toxin (CTX β , **Fig.1a**), and examined the distribution of fluorescently labelled neurons in vH. We made transverse slices of the vH to maintain the trisynaptic circuit layout, and to more easily identify the distribution of neurons across the different hippocampal subfields. Consistent with previous work, labelled neurons in these slices were located as a distribution, spread across the extent of the CA1/subiculum border (**Fig.1b**). However, in contrast to the graded distribution across the proximal:distal axis, we noticed that the distribution of labelled neurons was not uniform within the pyramidal layers, and consistently formed two distinct clusters distributed across the radial axis (equivalent to the most superficial and deep regions of the pyramidal cell layer (**Fig.1b,c**)).

We next confirmed this observation using unsupervised Gaussian Mixture Modelling (GMM), where the spatial distribution of vH-PFC neurons was consistently best described as two layers separated across the radial axis (**Fig.1d,e**). Together this suggests that the vH-PFC projection is distributed as a continuum across the CA1 subiculum border, but is segregated along the radial axis of vH.

Previously, the distribution of projection neurons in the hippocampus has been defined using standardized atlases such as the Allen Brain Atlas and the Hippocampal Gene Expression Atlas (HGEA). We next wanted to compare the distribution of vH-PFC neurons to these standard atlases. Therefore, we repeated our retrograde labelling experiment, prepared coronal sections of the entire extent of the hippocampus, and registered the location of labelled neurons to the Allen Brain Atlas (ABA) coordinate framework (**Fig.1f,g**). This allowed us to quantify the location of vH-PFC neurons across all three anatomical axes in standard reference space, and also their relative location in either CA1 and subiculum. Notably, and consistent with a gradient-like organization of vH output, labelled neurons were distributed across the entire area around the border of CA1 and proximal subiculum as defined by the ABA, with no obvious separation between the two regions (**Fig.1g, Sup.Fig.1a**). This was such that approximately half of the labelled neurons were present in both CA1 and subiculum (**Fig.1h**). However, it should be noted that the border between CA1 and subiculum in vH is contentious and can be dramatically different across studies and atlases (see discussion and **Sup.Fig.1**).

Consistent with our transverse sections, we noticed that in many coronal sections the distribution of labelled neurons was not uniform within the pyramidal layers, and again formed two distinct clusters distributed across the radial axis (**Fig.1g**). Previously, the distribution of projection neurons in the subiculum have been classified into distinct layers as part of the HGEA. We therefore aligned registered vH-PFC neurons to this atlas to investigate if the two vH-PFC populations may align with

these previously defined layers (**Fig.1g, Sup.Fig.1a**). We found that in subiculum the location of the two layers of vH-PFC neurons did not precisely map onto the layers defined in the HGEA. While the superficial layer corresponded to layer II, the deep layer was located at the border of layer III and IV. These results suggest that vH-PFC neurons are spread proximo:distally across the distal CA1/proximal subiculum border, but form two layers across the radial axis, that in subiculum correspond to superficial (layer II) and deep (layer III/IV) subiculum layers.”

and also added a substantial new section to the discussion starting on *p16, line 10*.

2- Authors use Calb-cre mice to target superficial CA1 cells. While this is the case for the dorsal hippocampus, it is unclear how calb expression relates to superficial CA1 cells at more ventral regions. It is also unclear how calb is expressed in the subiculum. In addition, authors should provide validation of their Calb-Cre on/off strategy. Finally, calbindin is also expressed in interneurons and in dentate granule cells and authors need to exclude any potential contribution/contamination. For interneurons in particular, calbindin expressing interneurons may be projecting cells. This issue is relevant as many of the viral strategies uses ubiquitous promoters. Thus, overall, the paper lacks detailed anatomical validation and support.

We agree with the reviewer and have now included detailed validation of our viral and genetic approaches as outlined below:

First, we have carried out immunostaining of calbindin combined with retrograde cholera toxin labelling from PFC to show that i) there is a graded distribution of calbindin labelling in ventral CA1/subiculum that is concentrated in the superficial layers, and ii) that this distribution overlaps with superficial layer vHPFC neurons labelled with cholera toxin. This data is shown in **Sup.Fig.3** and discussed on *p6, line 12*.

“We first performed a control analysis where we confirmed the selective expression of calbindin in superficial layers at the CA1/subiculum border in vH (**Sup. Fig.3a,b**), and that this superficial expression overlapped with superficial layer vH-PFC projection neurons labelled with CTX β .”

Second, we used a retrograde fluorescent ‘switcher’ virus. This expresses GFP in cre positive neurons and dTomato in cre negative neurons (Saunders et al, 2012). We injected this virus into the PFC of Calb1-IRES-cre mice, and examined labelling in vH. Consistent with our immunofluorescence above, we found that the majority of superficial neurons were GFP +, showing that they were Calb1 +ve, and the majority of deep neurons were dTomato +ve, showing that they were Calb1 -ve (An example is shown in **Fig.1i-k**, and this is quantified across animals in **Sup.Fig.3**). This is discussed on *p6 line 15*:

“Next, we injected a retrograde AAV expressing a fluorescence switch cassette into the PFC of a Calb1-IRES-cre mouse line. In this experiment, the color of the fluorophore expressed depended on the presence of cre (**Fig.1i-k**). Neurons projecting to PFC, that also express Calb1 (and therefore cre) will be labeled with one fluorophore, while neurons projecting to PFC that do not express Calb1 will be labeled with a different fluorophore. The expression of Calb1 robustly differentiated each population (**Fig.1i**), where the majority of superficial layer vH-PFC neurons expressed Calb1, while the majority of deep layer neurons did not (**Sup.Fig3c-e**).”

Third, we have characterised each of the Chr2, GCaMP and RGeCO1a viruses we used in the manuscript in more detail, again showing for each that Calb1 expression allows preferential expression in superficial layers using creON cassettes and deep layers using creOFF cassettes. This data is presented in **Sup.Fig.8** and **Fig.5**.

Fourth, we agree that long range inhibitory projections are an extremely interesting and important part of the hippocampal circuit, and that this must be controlled for. Thus, we have carried out two complementary experiments to investigate this. First, we retrogradely labelled PFC-projecting neurons using a cholera toxin injection in the PFC of a VGAT-cre mouse (VGAT is expressed in neurons utilizing the inhibitory transmitter GABA). We combined this with an injection of the cre-dependent tracer DIO-smFP-FLAG in vH. Thus, PFC

projecting neurons will be fluorescently labelled with cholera toxin, inhibitory interneurons will be labelled with smFP, and any putative long-range inhibitory projections will be dual labelled. Using this approach we found essentially no overlap between neurons labelled with cholera toxin and smFP, suggesting that long range inhibitory projections are not present.

To complement this retrograde analysis we also carried out an anterograde analysis where we compared axonal innervation of PFC from a viral injection into vH. We did this either from only excitatory vH neurons (using a CaMKii promoter) or only inhibitory neurons (using a cre-dependent tracer in a VGAT-cre mouse). Consistent with our retrograde experiment, we found that while excitatory input from vH to PFC was widespread, we could not find any axonal innervation of PFC from VGAT +ve vH neurons. These data are presented in **Sup.Fig.8b,c** and on *p12, line 2*.

“We first confirmed that there was no contamination of Calb1 creON innervation from putative long-range inhibitory projections that may also express calbindin by investigating if any PFC projecting neurons were positive for the inhibitory marker VGAT (**Sup.Fig.8b,c**).”

3- Authors use fiber photometry to evaluate activity of PFC-projecting deep and superficial cells during EPM exploration. While most analysis concentrate in evaluating transient events, the basal level of optical signal is not discussed nor shown. Only 4 sec of entry on the arm are analyzed so, it is unclear what is the typical activity along the entry and how does it evolve over exploration. Raw data and validation of a stable basal level should be provided as supplementary information. It is also unclear whether the fiber is sensing bulk activity from dispersed soma (as suggested in previous figures) and how precisely can deep and superficial cells be separated given their different density (and hence contribution to bulk imaging). Are fibers of passage or dendrites of deeper cells contaminating superficial signals? In addition, how can authors precisely define the entry of mice into the arm? All these experiments require substantial support from the anatomical and analytic point of views.

We thank the reviewer for their comments, and have fully reanalyzed our photometry experiments to address the reviewers concerns as outlined below:

First, we in hindsight realise we may not have explained our approach clearly in our original submission. Therefore, we have provided an extra figure (**Sup.Fig.9**) outlining the method for our photometry acquisition and analysis in more detail, providing examples of each step. In addition, we have expanded the methods on *p 31* to more clearly define our approach, and added extra histology images to the main figure (**Fig.5**) to make our approach clearer. In particular we have highlighted that we used a modulation:demodulation approach to acquire calcium sensitive fluorescence of the two calcium sensors we expressed in vH-PFC neurons (GCaMP6f in deep layer, Calb1-ve neurons, and RGeCO1a in superficial Calb1+ve neurons). Thus, we simultaneously recorded from a red calcium sensor expressed in superficial cells, and a green sensor expressed in deep cells. Thus, the separation of signals from the two layers is not based on anatomical separation, but upon Calb1 expression in a manner similar to experiments in previous figures. We hope with these extra additions, our approach is clearer, and we apologise for the confusion in our original submission. We have provided a more substantial explanation of our approach on *p 13, line 1*:

“To do this, we simultaneously recorded the activity of both populations using bulk calcium imaging through an optical fiber (**Fig.5a-d**). We used an injection of a mixture of retrograde AAVs into PFC to express both creON RGeCO1a – a red wavelength calcium indicator, and creOFF GCaMP6f – a green wavelength calcium indicator - in Calb1-IRES-cre mice. Using this technique, Calb1+, superficial neurons in vH projecting to PFC will express RGeCO1a, and Calb1- deep neurons projecting to PFC will express GCaMP6f. By collecting green and red fluorescence through an implanted optical fiber (**Sup.Fig.9**), this allowed simultaneous monitoring of both superficial and deep layer vH-PFC neuron activity in freely behaving mice while they explore the EPM.”

Second, as suggested by reviewers 1 and 3, we have carried out a more detailed analysis in order to investigate how activity of each layer changes during a number of key behavioral events throughout the session (**Fig.5, Sup.Fig.10**). This new analysis allows us to gain a more accurate picture of the time course of activity changes

during open and closed arm entry, and also allows us to include new data on the activity surrounding stretch attend, head dip, and rearing events. Interestingly we found no consistent changes in activity across these behaviors. This is described on *p13, line 9*.

“We aligned these recordings to the onset of different behaviors exhibited by mice as they explored the maze. Interestingly, activity surrounding rearing, stretch attend and head dipping behavior – classically seen as investigative behavior during maze exploration – was not consistently modulated (**Sup.Fig.10**).”

Third, in our original submission we presented transient events as these were the most consistent changes in activity we observed during the behavioural task, and this is common practice when using photometry to measure bulk activity (for example see Hung et al, 2017, Science; Kohl et al, 2018, Nature). However, in hindsight we agree with the reviewer that it would be helpful to provide a more holistic idea of how the activity unfolds across time, and across different behavioural events. Therefore we have presented the data in a number of extra ways to aid this, as outlined below:

i) We have provided an example of several minutes of recording during the task in the main figure (**Fig.5**). This gives an idea of baseline activity, and also suggests that – as would be expected –vH-PFC neurons are encoding much more than simply experimentally defined arm entries and behavioral events. We discuss this on *p19, line 18* where we suggest that a key future direction will be to use more highly quantifiable behavioral tasks to tease apart the role of these layers more thoroughly:

“On this note it is important to mention that activity of both layers of vH-PFC neurons was not limited to arm entries (**Fig.5d**) ... suggesting many more variables are represented by these neurons than we describe here. Therefore, an important future direction will be to investigate the activity of the two vH-PFC layers in behavioral tasks where the learning and updating of different stages of the decision process can be accurately separated.”

ii) We have presented our reanalyzed photometry data in **Fig.5** with an expanded time course, now looking at a 12 s epoch around the behavioral event. This has allowed us to observe both the overall baseline of the data, but also some interesting extra aspects of the activity of these neurons. For example, at the suggestion of reviewer 3, this has allowed us to examine the activity of the neurons both just before the behavioural event, and also after. Together, this suggests that in fact the majority of the differences between the two layers in vH occur after entry into the arm. This is presented on *p13, line 14*:

“Interestingly this activity was very different across the two layers of vH-PFC neurons. We found that upon exploration of the open arms of the maze, superficial neurons remained at baseline prior to arm entry, but transiently increased activity in the period after entry into the arm. In contrast, activity of deep layer neurons increased before entry to the open arm, but this activity was reduced dramatically upon arm entry (**Fig.5e,f**). Thus upon entry to the open arm, the superficial and deep layers undergo opposing changes in activity, consistent with our circuit mapping data.

Interestingly these changes in activity were also distinct when mice entered the closed arms of the maze. Superficial layer neurons exhibited no change in activity either before or after entry of the closed arm. Similar to open arm entry, deep layer neurons again increased activity prior to arm entry, but in contrast, this activity remained high when the mouse entered the arm (**Fig.5g,h**).”

We also discuss the potential consequences of this timing in a new section of discussion starting on *p18, line 21*.

iii) We investigated how activity of the two populations of vH-PFC neurons evolves from the entry of the arm until the mouse reaches the end of the arm (**Sup.Fig.11a**). We aligned the activity of each layer around both arm entry and arm end as 20 s epochs, and looked at how activity evolved across the arm. Consistent with our original description, activity of both populations was only transiently altered around the arm entry, and

quickly returns to baseline such that activity across the two layers is not consistently different by the time the mouse reaches the end of the arm. This is now discussed on *p14, line 11*:

“These changes in activity were transient, and activity returned to baseline levels by the time the animal had reached the end of arm exploration and turned to return (**Sup.Fig.11a**).”

iv) On the suggestion of reviewer 3 we have now also included an analysis where we examine how activity in each population changes across consecutive entries to the same arm. We found that while deep layer activity was consistent across consecutive entries, superficial layer activity was most pronounced on the first entry, and gradually decreased across successive arm entries. This is presented on *p14, line 12*:

“In addition, changes in superficial layer activity were much more prominent on the first entry to the open arm, and reduced over successive entries. In contrast, deep layer activity remained relatively stable across the session (**Sup.Fig.11b,c**).”

We also discuss on *p19, line 18*, that a key future direction will be to use more complex, quantifiable tasks to tease apart these roles, and how approach and avoidance are learnt and updated more effectively.

Fourth, overall, we feel it is important to discuss the limitations of the photometry approach explicitly – for example the extent to which bulk activity can reflect the activity of individual neurons. Therefore, we have added a section to our discussion *on p 19, line 8* to address this, where we conclude it will be an important future direction to understand the activity of these two populations at the level of single neurons.

4- Authors use optogenetics to manipulate the activity of deep and superficial PFC-projecting cells while mice explores the EPM. I am not fully convinced the behavioral effects are clean and statistically strong. First, it is unclear how the protocol goes. Authors light on only during the 3 min period in the central area or in the open arms. what if an animal stays in the closed arm? What proportion of animals receive lights actually? Second, there is lot of variability both in Cre-ON and GFP-control animals regarding the time in open arm, so it is unclear what the normalized time is doing here to reach significant statistical effects. All animals exhibit a decrease of exploration to the open arm (and hence they should be increasing exploration of the closed arm) along the task except for the CreON group, and some individual GFP-control mice. Thus, the effect of optogenetic looks a bit at the limit. Authors need to show convincingly that the behavioral effect is robust. Is stim in the closed arm exerting a complimentary effect that could reinforce statistics and conclusions?

We thank the reviewer for their comments and have performed additional analysis and experiments to address these concerns as outlined below:

First, we have reworded the text in both the results *p14, line 30*, and the methods *p 31*, and also made included an explicit schematic as part of the main figure (**Fig.6c**) to make it clear how our protocol is performed:

“Consistent with the transient nature of our photometry data, multiple previous studies have shown that transient manipulation of the vH-PFC circuit only around the center point and open arms is crucial for defining behavior. Therefore, we investigated whether artificially activating either the superficial or deep vH-PFC populations in this manner would influence mouse behavior on the maze. After a 3 min baseline, we artificially activated axons with blue light for a three-minute epoch, only when mice entered the central choice point in the EPM, and maintained excitation until the mice returned to the closed arms (**Fig.6c**). Thus, the activity of each population was only transiently artificially activated based on the mouses movement.”

In addition, a pre-experimental condition was defined such that if any mouse did not enter the centre point during the 3 min stimulation epoch it would be discounted. However, all mice that were tested passed this criterion and so were included in the final analysis. This is now explicitly described in the methods on *p 31*.

Second, we agree with the reviewer that it is unfortunate that there is a trend for all three groups to decrease exploration over time in the maze. This is unfortunately extremely common in the EPM (for example see Felix-Ortiz, et al, 2013, Neuron; Kim et al, 2013, Nature; Padilla-Coreano et al, 2016, Neuron). Therefore, to address the reviewer's concerns, we have performed two additional analyses. Together these new analyses complement the normalised data presented in the original manuscript:

i) In addition to the Mixed effects ANOVA showing that there is an interaction between group and epoch, we have now also carried out posthoc tests on the raw data (**Fig.6e**). We show that creON ChR2 animals explore the open arm more, and creOFF ChR2 animals less, when compared to GFP controls, only during the light on epoch.

ii) We quantified the probability of mice entering the open arms after entering the center zone of the maze (**Sup.Fig.12**). Again, this analysis revealed that creON mice had a higher probability of entering the open arms, while creOFF mice had a lower probability of entering the open arms, both compared to GFP controls, only during the light epoch. This data is presented on *p 15, line 3*.

“In addition, this manipulation altered the probability of entering the open arms in a similar fashion, where creON mice had a higher probability of entering the open arms during the light epoch, and creOFF had a lower probability, when compared to GFP controls (**Sup.Fig.12**).”

Third, we have carried out a new experiment (**Sup.Fig.13**) where we explicitly asked if light stimulation in the entire maze during the stimulation epoch altered the outcome of the experiment. We repeated our experiment in a new cohort of mice, but this time stimulated consistently across the entire 3 min stimulation epoch, similar to previous studies (Felix-Ortiz, et al, 2013, Neuron; Kim et al, 2013, Nature). Interestingly, in contrast to transient stimulation, we found that constant stimulation had very little effect on behavior in the EPM, similar to previous studies investigating this circuit (e.g. Jimenez et al, 2018, Neuron; Lee et al, 2019, Neuron; Padilla-Coreano et al, 2019, Neuron). Thus, it appears that transient, not constant activation of deep and superficial layers around the choice point of the EPM influences arm entry. This is presented on *p15, line 7*.

“We next asked whether this behavioral effect could also be seen with constant light stimulation during the 3 min light epoch – stimulation that was not dependent on the position of the mouse. Interestingly, and consistent with previous studies, this manipulation had minimal effect on behavior, suggesting that the transient activation of each pathway around the choice point in the EPM is necessary for changes in behavior (**Sup.Fig.13**).”

Other comments

5- Regarding quantification in Fig.1:

A) Please clarify whether quantification of radial distance in Fig.1b refers to several sections or is just one representative section;

We apologise for this not being clear in the original manuscript – this is a representative example, and was further quantified using Gaussian mixture modelling in the supplementary figures. In hindsight we realise it is better to include this in the main figures, and so now **Fig.1** contains this example, but also the quantification of all sections showing that vH-PFC neurons are consistently best described as two clusters. This result is discussed on *p4, line 19*.

“We next confirmed this observation using unsupervised Gaussian Mixture Modelling (GMM), where the spatial distribution of vH-PFC neurons were consistently best described as two layers separated across the radial axis (**Fig.1d,e**). Together this suggests that the vH-PFC projection is distributed as a continuum across the CA1 subiculum border, but is segregated along the radial axis of vH.”

B) Please, clarify whether cells are counted in a stack (70 μm thick) or at a plane; how many planes? What objective was used to quantify?

Cells throughout the manuscript were counted from sections obtained using wide field imaging of 70 μm slices. A minimum of 3 x 70 μm slices (planes) were used per injection for quantification. We used a Zeiss 10 x air immersion lens to image the slices. We have included this explicitly in the methods on *p27*.

C) In methods authors refer to “either the transverse, coronal or sagittal planes as described in the figure legends”, but most examples look horizontal or sagittal. No coronal sections are presented which would facilitate clarification of my point 1 (HGEA are represented in coronal sections).

We apologise for the confusion. As described above we have now included a new analysis using coronal slices and we agree this adds substantially to the paper and discussion.

D) Fig.1c and Supp.Fig.1d: authors refer to pia, but strictly speaking, this is the alveus. The pia is the layer of meninges surrounding the surface of the brain, which is more likely to be at the hippocampal fissure.

We have corrected this and apologise for the error.

E) Fig.1i, please improve contrast of the fluorescent image

We have increased the contrast as requested.

6- Check arrowheads in sup.Fig.2b; some point erroneously to distinct cells.

We have corrected this and apologise for the error.

7- Fig.2f: the retrobead injection is lacking. According to the text retrobead-negative cells are recorded here

We have corrected this and apologise for the error.

8- Authors declare “Excitatory local connectivity in hippocampal CA1 and subiculum is rare” but this is not strictly correct. Not certainly at the ventral subiculum, nor between CA1 and subicular cells.

We have corrected this and apologise for the error, and now refer only to local interneurons in this paragraph on *p8 line 26*.

9- Experiments in Fig.2g,h,i exploits somatic ChR2 but no supporting material is providing to validate their resolution. In addition, can authors fully exclude the fibers of passage? Also validation of interneuron-specific expression of mRuby (and electrophysiological validation) should be provided. A supplementary figure is critical here

In hindsight we agree with the reviewer that we did not present our characterisation of somatic ChR2 effectively. We have now made a new figure (**Sup.Fig.7**) showing recordings from soCoChR2 expressing and non-expressing cells in the presence of NBQX, APV and Gabazine to block synaptic transmission. We use focal light to probe the ability to fire action potentials in these two populations with light focussed on the soma, and 50 or 100 μm from the soma in either direction along the superficial to deep axis. We found that action potentials could only be reliably fired with the light focussed on the soma, and were almost completely absent 50 μm away. This is now explicitly discussed in the main text on *p9, line 10*.

“Next, we ensured that soCoChR2 expression allowed for light induced action potential firing with an appropriate spatial resolution. We recorded from soCoChR2 expressing neurons in the presence of synaptic blockers, and used a focused light spot to activate soCoChR2 in 50 μm increments centered around the soma of the recorded neuron. Consistent with previous reports we found that the ability to fire action potentials was limited to within 50 μm (**Sup.Fig.7**). In a separate experiment we also confirmed that soCoChR2 negative cells were not light sensitive.”

We also agree it would be helpful to validate the *dlx* promoter used for these experiments in vH. This promoter has been extensively characterised previously (see e.g. Dimidstein et al, 2016, Nature Neuroscience; Cho et al, 2015, Neuron). Therefore, we have carried out two new experiments to confirm this in our hands:

i) We injected a virus to express a fluorescent reporter under the control of the *dlx* promoter in the hippocampus of a *VGAT-cre* mouse. In the same animal we also injected the cre-dependent tracer DIO-smFP-FLAG to label *VGAT+* neurons in vH. As shown in **Sup.Fig.6a** there was extensive overlap between these two injections showing that the *dlx* promoter preferentially infects *VGAT +ve* neurons.

ii) We performed fluorescently guided whole-cell current clamp recordings from *VGAT-cre* mice injected with DIO mCherry (to label *VGAT+* interneurons) and in separate *VGAT-cre* mice compared the intrinsic properties of these neurons to fluorescently guided recordings from interneurons labelled with an injection of a virus expressing *dlx-mRuby*. As can be seen in **Sup.Fig.6b,c** – both labelling techniques allowed recordings from similar distributions of fast spiking and non-fast spiking hippocampal neurons, again suggesting that the *dlx* promoter can be used to target interneurons in vH. This is presented on *p9, line 9*.

“We first confirmed that the *dlx* promoter was a suitable tool to identify interneurons in vH (**Sup.Fig.6**).”

10- For the quantification of PFC-projecting cells in Fig.3a,b: it is unclear why authors present data as proportion of cells in the superficial layers and not along the normalized distance as they reported before. Interpreting this plot is very tricky. Also, in the table author refer the n for deep and sup which unclearly relate to CamKII vs *VGAT* (as shown in Fig.3b).

We apologise for this not being clear, and for the error in the labelling of our statistics table. We have now updated our description of the experiment, and the statistics table to make this clearer, as well as added a new panel to the main figure (**Fig.4e**). Importantly, and what was not clear in our original manuscript, is that the distribution of both cholera toxin-labelled and rabies-labelled neurons across superficial and deep layers utilized here is based on the clustering presented in **Fig.1d,e** (originally **Sup.Fig.1**). Using unbiased Gaussian Mixture Modelling, all labelled neurons in a slice were split into two groups (corresponding to the superficial and deep layers) based on their spatial distribution in the slice. We then compare the number of rabies labelled neurons in each automatically defined layer. After this classification, the proportion of rabies-labelled cells in the superficial layer (presented in **Fig.4e**) can be calculated. This is now explicitly discussed on *p10, line 34*:

“We then split labelled neurons into two layers using the unbiased GMM approach outlined in **Fig.1**. Following GMM analysis, each rabies labelled neuron was classified as being in either the superficial or deep cluster (**Fig.4e**) and the proportion of rabies labelled neurons in each layer for either *VGAT* or *CaMKii* starter cells in PFC could be calculated.”

11- Regarding axonal arborization of PFC-projecting cells in Fig.3c, authors need to show the distribution of the axonal plexus in the PFC to clarify whether deep and sup cells are actually innervating different layers and/or dendritic domains which could additionally complicate interpretation.

We have now carried out a new set of axonal tracing experiments presented in **Fig4a,b** comparing the distribution of creON and creOFF vH input into PFC. These experiments show that across all three anatomical axes (ML, AP, DV) there are no obvious differences in the distribution of axons. Thus, differences observed are likely mainly to be due to unique postsynaptic targets rather than spatial distribution of presynaptic terminals. This finding is discussed on *p10, line 18*.

“We investigated whether superficial and deep layer vH-PFC neurons innervated unique spatial locations within PFC ... We then made coronal slices of PFC from these animals to investigate the distribution of axons from these neurons. We found that there were no obvious differences in the distribution of axons from the two layers across the AP, ML or DV axis (**Fig.4.a,b**). This suggests that the two layers of vH-PFC neurons, despite receiving distinct afferent input, innervate similar areas and layers of PFC.”

Interestingly, authors record only from deep PFC cells, while Lee et al., Neuron 2019 (ref 44) record from layers II/III. Fig.3f: what PFC layers are recorded here? All this should be more carefully considered and justified.

We chose to record from layer V neurons in PFC as previous studies have shown that the majority of input from vH arrives into these layers (e.g. Liu et al, 2018, J Neuroscience), and anatomical data from our study (**Fig.4a,b**) and others suggests that axon density from vH is highest in deeper layers. We have now presented this reasoning on *p12, line 10*.

“We chose to record from deep, layer V neurons in PFC as our anatomy and that from previous studies have shown that the majority of input from vH arrives into these layers.”

However, more superficial layer II/III neurons also receive input from vH, and although sparse, there are a number of vH axons in superficial layers of PFC. It is interesting to note that the properties of vH input have been suggested to change across the layers of PFC, as well as across principal neurons with different output projections and so we have now highlighted on *p18, line 12* that an interesting future direction would be to compare the influence of each vH layer across the different layers and neuron types of the PFC.

12- Supp.Fig.3: the injection site and fiber placement: please, show some examples to appreciate the size and potential anatomical effect of the fiber implantation.

We have now added examples for both photometry and optogenetics figures (**Fig.5, Fig.6, Sup.Fig.9**).

13- I strongly recommend separating fiber photometry and optogenetic behavioral experiments in different figures, and adding substantially more analysis, and support to results.

We agree and have now separated these two experiments into separate figures (**Fig.5, Fig.6**), and as detailed above (responses to the reviewers points 3 and 4) added substantially more analysis to both figures.

14- Depending on their ability to address my point 1 I would recommend authors making more clear the CA1/subicular nature of the PCF-projecting cells and shape title and abstract accordingly.

We have updated the abstract and main text substantially to include more discussion about the identity of these neurons. However, as the exact identity of CA1/subiculum in ventral hippocampus is still in discrepancy, we have kept the title. We will be happy however to make further changes to the title as the reviewer sees fit.

15- Discussion, in general reads a bit oversimplified. I would have several comments and additions but in view of the overall assessment and the many details authors need to work out to improve this interesting paper I would save my comments for a revised version.

We have substantially reworked and added to the discussion, and tried to make pre-emptive changes, in a large part due to the interesting extra experiments and analysis requested by the reviewers. We hope this will now be more appropriate.

Reviewer #2 (Remarks to the Author):

In their manuscript, Sánchez-Bellot and MacAskill examine the local- and long-range connectivity of two circuits connecting the ventral hippocampus to the prefrontal cortex, which differ based on radial depth of ventral hippocampus neurons, and relate these circuits to approach/avoidance behavior. They associate activity in the deep ventral hippocampus circuit with avoidance behavior and activity in the superficial ventral hippocampus circuit with exploration/approach. Finally, they show that optogenetic activation of each of these circuits leads to the predicted behavior, demonstrating a causal mechanism by which the hippocampus plays a bidirectional role in approach/avoidance behavior.

The experiments presented in the manuscript are challenging and well-executed. The paper does an outstanding job of bridging current coarse scientific knowledge into fine-scale discovery: the authors hinge their experiments on previously identified cell types (i.e., superficial vs. deep ventral hippocampus neurons) and a known bidirectional role for the hippocampus in approach/avoidance, but are the first to show that this hippocampus function derives from distinct circuits involving these cell types. As such, the paper fills a clear knowledge gap and is timely. Given their novel findings spanning multiple scales (i.e., circuit mapping to behavior), recapitulation and extension of previous results, and the potential eventual translational relevance of their results, this manuscript is of interest to broad audience and well-suited to a strong journal like Nature Communications.

In this reviewer's opinion, no major revisions nor further experiments are needed. However, the clarity and convincingness of some parts of the paper could be improved by considering the following minor comments and addressing via additional discussion or rewording.

We thank the reviewer for their positive appraisal of our manuscript. We have updated the current version as outlined below to address the reviewers concerns.

Minor comments:

- CRACM-based conclusions: I would suggest that the authors exhibit more restraint in discussing their CRACM-associated conclusions for Fig. 2. Their recordings are not well-positioned to compare inputs across superficial vs. deep cells for at least two reasons. First, morphological differences will mean that inputs – especially distal inputs – will have differential somatic signatures. Second, recordings were performed in current clamp, which means that cell-intrinsic differences may distort synaptic inputs. Thus, more equivocal wording should be employed – claims like “additional input” can be suggested by not truly resolved by these experiments.

We agree with the reviewer and have altered our description of these experiments throughout the manuscript to couch our interpretation. In particular we have explicitly stated that we record only the influence of these inputs at the soma on *p8, line 14*:

“Using brief pulses of blue (473 nm) light allowed us to directly compare the relative influence of synaptic input at the soma of superficial or deep layer vH-PFC neurons from each of the afferent regions”

- Off-target effects, optogenetic stimulation: Stimulation of axon terminals can result in antidromic spiking back to the cell body, which can then orthodromically propagated to other axon terminals that are associated with different collateralized circuits. This off-target stimulation of collateralized circuitry can be a potential confound. The authors should discuss this possibility in the context of their experimental results.

We agree with the reviewer, and a similar point was also raised by reviewer 3. To address this, we have now carried out an extra experiment investigating the collateralisation of output across the superficial and deep layers of vH-PFC neurons (**Sup.Fig.2**). Interestingly we find that the majority of neurons with dual projections to both PFC and either NAc or LH are located in the deep layer. This is presented on *p6, line 1*:

“... we injected fluorescent cholera toxin into either PFC and nucleus accumbens (NAc) or PFC and lateral hypothalamus (LH). First, we investigated overall colocalization of tracers – which is

suggestive of collateral projections to both regions (**Sup.Fig.2a,b**). Consistent with previous studies we found that for both NAc and LH experiments only a small percentage of neurons were labelled with both tracers, suggesting they sent collaterals to both regions. We next looked at how these dual-projecting neurons distributed across the two layers of vH-PFC neurons. Interestingly we found that of all dual-labelled neurons, almost 80% were in the deep layers, with very few in the superficial layer (**Sup.Fig.2c**). Thus, vH neurons that project to PFC have only very limited collateralization, but those neurons with collaterals are preferentially found in the deep, not superficial layer.”

Importantly, we also discuss the potential biological and technical consequences of this result in the main text on *p19, line 31*.

“ ... terminal stimulation using optogenetics is not physiological, and can cause antidromic spiking. This could result in activity in areas with collateral projections from vH-PFC neurons such as NAc and LH. Although neurons with such collaterals were relatively rare in vH (~5%, **Sup.Fig.2**), we found that these neurons were found preferentially in the deep layers. Interestingly, antidromic spiking in this projection is thought to be very rare (Padilla-Coreano 2019), however, it is unknown what effect such collateral projections might play.”

• CA1 vs subiculum: While the authors state that the neurons they study are at the “CA1/subiculum” border, they do not attempt to describe which brain region (CA1 vs subiculum) these neurons belong to. Given that the exact boundary between the CA1 and subiculum has been debated in the field (especially in ventral hippocampus, as in this study), this is a challenging task. Nonetheless, some discussion as to what hippocampal region these cells reflect, or indeed if this is even possible to interpret the context of the authors' experiments, is important. As many in the community think about these outputs in terms of CA1 vs. subiculum, it will be important for this paper to discuss their results in the context of these regions.

In hindsight, and as also suggested by reviewer 1, we agree that a more detailed analysis and description of the location of vH-PFC neurons within the CA1 and subicular subfields is important. Therefore, we have performed an additional experiment where we serially sectioned the entire hippocampus coronally and registered the location of cholera toxin labelled vH-PFC neurons to the Allen Brain Atlas coordinate space. This allowed us to investigate the precise distribution of vH-PFC neurons with respect to the previously defined anatomical borders of CA1 and subiculum, but also to align this data to the Hippocampal Gene Expression Atlas (Bienkowski et al 2018) to investigate whether the two layers of vH-PFC neurons correspond to the previously defined three-layer structure of ventral subiculum.

In summary, we find that in contrast to dorsal hippocampus where the boundaries of e.g. CA1, proximal and distal subiculum are relatively sharp and well defined, the distribution of neurons in ventral hippocampus is much more of a gradient. This results in vH-PFC neurons being evenly spread as a continuum across both CA1 and subiculum, where the most posterior parts of this distribution align with previously described superficial and deep layers of subiculum (Bienkowski et al 2018, Ishihara and Fukuda, 2016). We present this data in a new results section starting on *p4*, and have also added a substantial new discussion section, where we note that investigating the differences between these two regions in the future is very important, on *p16, line 16*:

“... it is becoming increasingly evident that in contrast to dorsal subiculum, which is arranged in a tight, spatial distribution with sharp borders between subfields, ventral hippocampal subfields seem to be organized as a gradient, with few distinct anatomical delineations between subfields. Therefore, in our study we have treated vH-PFC neurons as a continuum that is consistently segregated into two populations across the radial axis. However, it is important to note that the difficulty in defining the CA1 and subiculum border in ventral hippocampus means that our study did not explicitly differentiate these two distinct subregions of vH. It is known that these two subfields perform distinct roles in the hippocampal calculation, and have distinct local and long-range connectivity. Thus, our data cannot differentiate this important information. Future work – potentially by using multiple levels of intersectional targeting – is needed to more accurately define the borders between subfields in vH, and to tease apart the differential contribution of superficial and deep CA1-PFC and subiculum-PFC connections.”

- CreON vs. creOFF explanation: The wording describing the creON/creOFF experiment was confusing and seemed contradictory: “[We] injected an AAV into vH to express either ChR2 where expression was limited to cre-expressing (superficial layer) neurons (creON ChR2), or where ChR2 was inhibited in cre-expressing neurons, and thus was only expressed in cre-negative (deep layer) neurons,” as the sentence structure makes it seem as though in both creON and creOFF ChR2 is expressed in only cre-negative neurons.

We apologise that this wording was unclear. We have now updated this sentence to make it clearer, on *p10, line 19*.

“.. we injected an AAV into the vH of Calb1-IRES-cre mice to express either ChR2 where expression was restricted only to cre-expressing (superficial layer) neurons (creON ChR2). In a separate experiment we injected an AAV where ChR2 was inhibited in cre-expressing neurons, and thus was only expressed in cre-negative (deep layer) neurons (creOFF ChR2, Sup.Fig.8a).”

- Figure legends: It would be helpful to have a figure legend that describes what the orange neuron in Figure 3c is (it is provided in 3f, but would make more sense to have this legend in the first panel the reader views). In the current text, is also unclear how this schematized neuron is related to the experiment.

We have updated both the text and the figure to make this clearer.

- Reporting of cell sample sizes: Often, the number of superficial vs. deep ventral hippocampus and PFC analyzed cells are not reported in the text or on relevant figures (e.g., Figure 1g– percentage of bursting cells is reported for both superficial and deep hippocampus cells, but not the actual number of cells examined; although there are dots for individual cells, the burden is on the reader to manually add the data points). Please report sample sizes throughout, as possible.

We apologise for this not being clear, and have updated the statistics table to make this as clear as possible. In addition we have now explicitly stated in the figure legend for **Fig.2** that this information is in the supplementary statistics table. However, if the reviewer would prefer we would be happy to move this information to the figure legends.

Reviewer #3 (Remarks to the Author):

Sanchez-Bellot & MacAskill used a combination of in vitro slice electrophysiology, anatomical tracing, in vivo fibre photometry and optogenetics to identify two populations of medial prefrontal cortex (mPFC)-projecting ventral hippocampal (vH) cells that have distinct electrophysiological and projection profiles, as well as opposing control over the expression of innate approach-avoidance conflict, as measured in the elevated plus maze (EPM). The major findings of the study are that vH cells in the superficial layers in the subiculum and CA1 receive medial entorhinal inputs, and connect to both somatic and dendritic inhibitory neurons in the PFC in order to facilitate approach/exploratory behavior. In contrast, vH cells residing in the deep layers receive subcortical projections (anterior thalamus, DBB) and project onto somatic interneurons and pyramidal cells in the PFC to promote avoidance behavior.

This is a very interesting set of experiments that have been conducted carefully and thoroughly. The findings provide novel mechanistic insights into the control of anxiety, and will likely be of interest to an audience interested in the neurobiology of anxiety, and function of hippocampal and prefrontal circuits. The statistical analyses are appropriate, and the methodology is clearly written and reproducible on the whole, although more clarification and details are warranted in some parts, as described below. I have also outlined some concerns that would need to be addressed in order for me to fully assess the validity of the claims.

We thank the reviewer for their very positive comments. We have carried out a number of extra experiments and analyses outlined below that we hope address the reviewers concerns.

I appreciate the reason why the authors chose to stimulate contralateral vCA3 inputs to the PFC projecting cells, but this raises the question of whether there may be intrinsic differences in the projection pattern/physiological profile of the neuronal population targeted by ipsilateral vs. contralateral vCA3 inputs, and whether the omission of recording from the ipsilateral pathway in the current study is an issue.

We agree with the reviewer, and have carried out experiments using electrical stimulation of ipsilateral CA3 input in **Sup.Fig.5e,f**. We found that this experiment mimicked that of contralateral input – in that there was no clear bias towards either population. As a control for this experiment, we also carried out electrical stimulation of the ipsilateral entorhinal cortex input via the temporoammonic pathway. Again, this experiment recapitulated our ChR2 results with a clear bias to superficial vH-PFC neurons. These experiments are now explicitly highlighted in the text on **p8, line 20**.

“We subsequently confirmed differences in local input from ipsilateral CA3 and ENT using electrical stimulation of the Schafer collaterals and temporoammonic pathway respectively (**Sup.Fig.5e,f**).”

Did the authors assess the collateral projections of the superficial and deep layer PFC-projecting vH cells?

In our original manuscript we did not carry out this experiment. However, we agree with the reviewer that this is an interesting and important point. Previous studies have shown that there is a small but consistent percentage of neurons in vH that project to multiple areas (e.g. Naber and Witter, 1998; Wee and MacAskill, 2020; Gergues et al, 2020). Therefore we performed a new experiment to investigate whether the two layers contained different proportions of these dual projecting neurons. We carried out a series of dual retrograde labelling experiments where we injected fluorescent cholera toxin in pairs of either PFC and Nucleus Accumbens (NAc), or PFC and Lateral Hypothalamus (LH). Interestingly although overall levels of collateralisation were very low, we found that almost 80% of this subpopulation of dual projecting neurons were located in the deep layers, with very few colocalised neurons in the superficial layer. These data are now presented on **p6, line 1**:

“... we injected fluorescent cholera toxin into either PFC and nucleus accumbens (NAc) or PFC and lateral hypothalamus (LH). First, we investigated overall colocalization of tracers – which is suggestive of collateral projections to both regions (**Sup.Fig.2a,b**). Consistent with previous studies we found that for both NAc and LH experiments only a small percentage of neurons were labelled with both tracers, suggesting they sent collaterals to both regions. We next looked at how these dual-projecting neurons distributed across the two layers of vH-PFC neurons. Interestingly we found that of all dual-labelled neurons, almost 80% were in the deep layers, with very few in the

superficial layer (**Sup.Fig.2c**). Thus, vH neurons that project to PFC have only very limited collateralization, but those neurons with collaterals are preferentially found in the deep, not superficial layer.”

Importantly, this difference in collateralisation may have important consequences both for the biological functioning of the two layers, but also technically for the interpretation of optogenetic experiments due to the potential for antidromic effects of stimulation. These important points are now discussed on *p19, line 31*:

“... terminal stimulation using optogenetics is not physiological, and can cause antidromic spiking. This could result in activity in areas with collateral projections from vH-PFC neurons such as NAc and LH. Although neurons with such collaterals were relatively rare in vH (~5%, **Sup.Fig.2**), we found that these neurons were found preferentially in the deep layers. Therefore, although antidromic spiking in this projection is thought to be rare (Padilla-Coreano 2019), it is unknown what effect such collateral projections might play.”

and *p18, line 35*:

“...deep layer activation may drive a more general state of avoidance or exploitation – consistent with its collateral projections to NAc and LH”

The behavioral and data analyses that the authors conducted are not best suited to justifying some of the claims the authors are making. For instance, the authors claim their fiber photometry findings “suggested that the relative activity of the superficial and deep layers of the vH-PFC projection around the choice point of the EPM may inform the decision to approach or avoid the open arms.” However, neural activity was analysed during 0 to 4 seconds after arm entry, and changes in activity also appear most prominent after the entry into either the open or closed arm (Figure 4 c,d). Given that approach and avoidance decisions likely occur before the animal enters the arm, and not after, it is possible that the time window the authors include in their analysis may not be representative of decision-making. Was neural activity analysed for the 2s preceding arm entry (-2s to 0s)?

We agree with reviewer that our original interpretation of the photometry data may have been overly simple. In combination with suggestions from reviewer 1 we have now carried out extra analysis that allows us to investigate this in more detail as outlined below:

As described in our response to reviewer 1, in **Fig.5**, and **Sup.Fig.10** we have now scored the behaviour in greater detail and presented the data in the figures as a longer, 12 s epoch around the behavioural event. This more easily allows us to investigate the changes in activity before and after arm entry as described by the reviewer. From this analysis it is clear there are multiple successive stages of activity during arm entry, which we describe on *p13, line 10*:

“... activity surrounding rearing, stretch attend and head dipping behavior – classically seen as investigative behavior during maze exploration – was not consistently modulated (**Sup.Fig.10**). However, we noticed consistent changes in activity in both layers when mice moved into either the closed arms or the open arm of the EPM, and that this activity was apparent both before entry into the arm, and after arm entry (**Fig.5e-h**). Interestingly this activity was very different across the two layers of vH-PFC neurons. We found that upon exploration of the open arms of the maze, superficial neurons remained at baseline prior to arm entry, but transiently increased activity in the period after entry into the arm. In contrast, activity of deep layer neurons increased before entry to the open arm, but this activity was reduced dramatically upon arm entry (**Fig.5e,f**). Thus, upon entry to the open arm, the superficial and deep layers undergo opposing changes in activity, consistent with our circuit mapping data.

Interestingly these changes in activity were also distinct when mice entered the closed arms of the maze. Superficial layer neurons exhibited no change in activity either before or after entry of the closed arm. Similar to open arm entry, deep layer neurons again increased activity prior to arm entry, but in contrast, this activity remained high when the mouse entered the arm (**Fig.5g,h**).

This new analysis however results in an important new point of discussion. Crucially, it is now clear that the activity of the superficial layer, and importantly also the appearance of differences between the activity of the superficial and deep vH-PFC layers, peak after entry into the arm, not preceding it. This is similar to previous reports of both vH and PFC activity during EPM exploration. We therefore agree with the reviewer that this suggests that our original interpretation of the photometry results requires updating. We now discuss this in a substantial new section of the discussion on *p18, line 21*:

“We found that the two layers of vH-PFC neurons exhibited opposing changes in their activity upon open and closed arm entry in the EPM (**Fig.5**). However, it is interesting to note that this activity was predominantly after entry to the arm, and hence potentially after the decision to enter the arm has been made. This is consistent with the majority of activity seen in PFC and vH during EPM exploration where specificity in neuronal activity is predominantly after arm entry. However, this does not immediately fit with the previously described key role for this circuit in the decision to enter the open or closed arms, and also our optogenetic data showing that activation of each layer could bidirectionally bias entry to either the open or closed arms (**Fig.6**).

One explanation for this discrepancy might be that vH-PFC neurons simply define whether the arm is open or closed once the mouse has entered the arm (or more generally whether the mouse is currently approaching or avoiding potential adversity. This fits well with the idea of the hippocampus providing information about ‘state’, with subsequent decisions based on this information carried out in PFC and other downstream regions. In this scenario, artificial activation of superficial vH-PFC activity may drive exploration or approach, while deep layer activation may drive a more general state of avoidance or exploitation – consistent with its collateral projections to NAc and LH ... [alternatively] this discrepancy could also be due to the inherent disadvantage of the way the classic EPM task is carried out - where decisions and internal variables are inferred by the location of the mouse within the maze. Due to this issue, it is difficult to precisely identify the timing of decision variables. For example, in the EPM it is presumed that the decision to enter the open arm is made just before entry, when in fact this choice could be made over a large period of time ranging from many seconds before arm entry to when the mouse is already in the proximal parts of the chosen arm. On this note it is important to note that activity of both layers of vH-PFC neurons was not limited to arm entries (**Fig.5d**), suggesting many more variables are represented by these neurons than we describe here. Therefore, an important future direction will be to investigate the activity of the two vH-PFC layers in behavioral tasks where the learning and updating of different stages of the decision process can be accurately separated.”

Clarification is needed as to how exactly the $\Delta F/F$ values were obtained. The methods state “Time series data was then calculated for each behavioral session as the z-score of (green or red signal - fitted autofluorescence signal)” which accounts for the the z-score of ΔF shown in the traces. The authors also state, “Activity was analyzed as the area under the curve from 0 to 4 s after arm entry, normalized to the area under the curve -4 to -2 s before arm entry”. I assume this resulted in the reported $\Delta F/F$ values but it is not clear how these were derived.

We apologise for this confusion, as our terminology of $\Delta F/F$ was not accurate. We have now made a new figure outlining the approach we took for our photometry recordings (**Sup.Fig.9**), have updated the figures to refer simply to zF , and have also updated the methods on *p31* to make this clear.

Please provide details as to how many open and closed arm entries were included in the data analysis. Did the observed changes in activity differ between successive entries into the open or closed arms (those made at the beginning vs. end of session)?

Animals underwent multiple arm entries (ranging from 1 to 13, 6.2 +/- 2.4 (mean +/- sem) open arm entries and 9 to 17, 14 +/- 0.9 closed arm entries). For the analysis in the main figure, **Fig.5** and **Sup.Fig.10** we focussed on the average signal around each behavioural event for each mouse. However, we agree that a fascinating question is how this signal develops across multiple open arm entries. Therefore in **Sup.Fig.11b,c** we have reanalysed our photometry data with respect to the first, middle and last arm entry. Interestingly we find that the

dynamics of the deep layer neurons are relatively consistent across the session. However, superficial layer activity was prominent on the first open arm entry, but was markedly reduced at the end of the session. This suggests that the superficial layer may be preferentially involved during the open arm entries early in each behavioural session. We present this data on *p14, line 13*.

“... changes in superficial layer activity were much more prominent on the first entry to the open arm, and reduced over successive entries. In contrast, deep layer activity remained relatively stable across the session (**Sup.Fig.11b,c**).”

We also have added a section to the discussion where we highlight that a behavioral task designed to be able to examine how approach and avoidance is learnt and updated over experience is an exciting future direction to address this more thoroughly on *p19, line 19*.

“... it is important to mention that ... the activity of vH-PFC neurons was not constant over consecutive arm entries (**Sup.Fig.10**), suggesting many more variables are represented by these neurons than we describe here. Therefore, an important future direction will be to investigate the activity of the two vH-PFC layers in behavioral tasks where the learning and updating of different stages of the decision process can be accurately separated.”

It is of concern that the placements of optic fibres in the fibre photometry experiments seem to be much more posterior and dorsal to the sites targeted in the tracing and in vitro slice experiments (Supp Fig 3b).

We apologise that this was not clear, our aim for the photometry experiments was to implant our excitation / emission fibres ~300 μm above the injection site, which optimises bulk signal (see e.g. Gunaydin et al, 2014, Cell). The injection sites, and subsequent expression of constructs for photometry experiments were the same coordinates for all other experiments. We have now explicitly stated this in the methods on *p26* and included histology images in the main figure **Fig.5** to make this clearer.

The inclusion of the optogenetic activation experiment is a nice way of demonstrating the sufficiency of the differential vH-mPFC circuits in approach/avoidance behaviors. However, it is not a demonstration of the necessity of these circuits in the control of anxiety – did the authors examine the effects of inhibiting these circuits on approach/avoidance?

We agree with the reviewer that inhibition data would be an excellent addition to this study. However, there are a large number of technical issues with current optogenetic tools – including large off target effects and spurious, often excitatory currents that make this experiment extremely challenging. Due to these limitations we feel that these experiments are beyond the scope of our current study. To address this, we have included a new section in the discussion where we outline these issues in detail. Most importantly we highlight the very recently released G-protein coupled optogenetic tools that may circumvent the majority of these issues in the future. This is presented on *p19, line 36*.

“... while our optogenetic experiments show that each pathway is sufficient to change behavior in the EPM, they do not show that the activity of these pathways are necessary to support such behavior. There are a number of caveats that make such experiments difficult to interpret. For example, terminal manipulation using the majority of inhibitory opsins have large, off-target effects that often result in opposite physiological effects than those intended, while inhibition using retrograde infection is often inefficient, and confounded by collateral projections to distinct downstream regions (see **Sup.Fig.2**). However, recently developed G-protein coupled optogenetic constructs - which lack the caveats of classic optogenetic tools when used at terminals - may provide a means to carry out these experiments for the first time. Therefore an important future direction is to investigate the necessity of the two pathways during approach avoidance conflict.”

Perhaps the manuscript is intended to be a short report, but I found the discussion to be lacking, and have a few suggestions for improvements. Firstly, the authors should discuss their findings in the light of recent work that

explore the contributions of different vH subfields in innate and cued approach-avoidance behaviors (e.g., PMID: 25931556, PMID: 29606418, PMID: 32929245). Secondly, it's not clear how exactly the reported circuit mechanisms (particularly the contrasting innervation patterns of the mPFC) give rise to the implementation of preferential approach vs. avoidance in a conflict situation. Furthermore, huge conceptual leaps are made in stating that superficial neurons use structured information from the cortex to plan and promote approach/exploratory behavior, and deep layer neurons use salient information in the environment flexibly to promote avoidance. The authors should better articulate/elaborate on how they have arrived at these conclusions.

We agree with the reviewer, and this was also pointed out by reviewer 1. We have substantially updated and added to the discussion, including discussing each of these papers. We hope that this new discussion will satisfy the reviewers concerns.

Please indicate if, and which within subject post-hoc comparisons were significant in the EPM data (Figure 4 g, i)

We thank the reviewer for pointing this out. Posthoc testing has now been added to the optogenetics dataset. These tests show that for the raw data, creON increases, and creOFF decreases open arm time only in the stimulation epoch compared to GFP controls. This data is now presented in **Fig.6**.

REVIEWERS' COMMENTS

Reviewer #1 (Remarks to the Author):

The authors have revised the manuscript thoroughly with new data and analyses that address all my previous comments. I would like to congratulate them for a serious work. The current version reads very clear and I have no additional major concern. While I still feel most of the data come from the prosubiculum I appreciate the authors' take and the balanced view they now provide. There are only some minor issues to revise:

1- Calbindin immunostaining is not described in methods (i.e. details regarding the protocol and primary/secondary antibodies)

2- Supp.Fig.2A: Since the image is flipped with respect to the previous one, it may be useful to indicate where is proximal and distal, as well as deep and superficial.

3- Page 10, Line 25: 'fromm'

4- Page 12, Line 2 'as before, Fig.4)'

Reviewer #3 (Remarks to the Author):

All my concerns have been addressed thoroughly, and I have no further comments.

Reviewer #1 (Remarks to the Author):

The authors have revised the manuscript thoroughly with new data and analyses that address all my previous comments. I would like to congratulate them for a serious work. The current version reads very clear and I have no additional major concern. While I still feel most of the data come from the prosubiculum I appreciate the authors' take and the balanced view they now provide. There are only some minor issues to revise:

We thank the reviewer for their previous suggestions the we agree have made the manuscript much stronger. We address the final comments below:

1- Calbindin immunostaining is not described in methods (i.e. details regarding the protocol and primary/secondary antibodies)

We apologise for this omission, we have now included a section in the methods on **p 19, line 42** of the new manuscript.

2- Supp.Fig.2A: Since the image is flipped with respect to the previous one, it may be useful to indicate where is proximal and distal, as well as deep and superficial.

We agree with the reviewer and have now updated **Supp.Fig.2a** to address this. First, we have reoriented the images so that they are now consistent with the remainder of the examples in the manuscript. Secondly, to aid clarity, we have included schematics of the injections sites as well as the location of the images in the hippocampal slice. This can be found on **p 3** of the new supplementary information file.

3- Page 10, Line 25: 'fromm'

Thank you for spotting this, we have corrected this on **p 9, line 21** of the new manuscript.

4- Page 12, Line 2 'as before, Fig.4)'

Thank you for spotting this, we have corrected this on **p 10, line 2** of the new manuscript.

Reviewer #3 (Remarks to the Author):

All my concerns have been addressed thoroughly, and I have no further comments.

We thank the reviewer for their previous comments.